# HPS: Hyperspherical Parameter Sharing for Efficient Multi-Agent Reinforcement Learning

**Hu Fu** [1 2]  **Pengyi Li** [3]  **Hao Chen** [4]  **Xuanyu Xiang** [1 2]  **Biao Luo** [5]  **Yihua Tan** [1 2]

## Abstract

Parameter Sharing (PS) is widely used to improve efficiency in Multi-Agent Reinforcement Learning (MARL), but it can limit behavioral diversity and degrade performance. This limitation stems from gradient conflicts among agents on shared weights, which hinders effective policy learning. To fully characterize this phenomenon, we propose Geometric Gradient Decomposition Analysis that decomposes gradients with respect to weight vector into radial (scale) and tangential (direction) components and uncover a key insight: agents largely agree on directional updates but substantially disagree on scale updates. Consequently, while recent methods split the shared network into agent-specific subnetworks to mitigate conflicts, they also discard shared directional updates, limiting training efficiency. To address this issue, we propose Hyperspherical Parameter Sharing (HPS), which explicitly decouples direction and scale in parameter sharing. Specifically, HPS constrains the shared backbone weights onto a Riemannian manifold(unit hypersphere), enforcing purely directional learning. Building on this, an agent-specific scale generator outputs multiplicative modulation factors to adjust each agent's scales, thus preserving heterogeneous response magnitudes without disrupting the shared directions. Experiments on SMAC, SMACv2, VMAS and Predator Prey demonstrate that HPS effectively resolves the scale conflict, significantly outperforming state-of-the-art methods.

---

[1]School of Artificial Intelligence and Automation, Huazhong University of Science and Technology(HUST), China [2]National Key Laboratory of Multispectral Information Intelligent Processing Technology, HUST, China [3]Tianjin University, China [4]Institute of Automation, Chinese Academy of Sciences, China [5]School of Automation, Central South University, China. Correspondence to: Yihua Tan <yhtan@hust.edu.cn>.

*Proceedings of the 43rd International Conference on Machine Learning*, Seoul, South Korea. PMLR 306, 2026. Copyright 2026 by the author(s).

## 1. Introduction

Cooperative multi-agent reinforcement learning (MARL) has gained significant attention lately due to its potential applications in a variety of real-world challenges, including traffic light management (Zhang & Lesser, 2011), autonomous vehicles (Singh et al., 2020), and swarm robotics (Hüttenrauch et al., 2017). A standard paradigm underlying these successes is Parameter Sharing (PS). By training a single unified policy network across all agents, PS maximizes sample efficiency and accelerates convergence (Zhang et al., 2021). However, it neglects the varying decision requirements across agents, leading to insufficient behavioral diversity and degrade performance

To overcome this problem, existing research has mainly investigated two approaches. The first approach focuses on explicit role modeling or subtask decomposition. ROMA (Wang et al., 2020a) and RODE (Wang et al., 2020b) introduce latent role concepts to decompose the complex joint action space, guiding agents toward role specialization. LDSA (Yang et al., 2022) and LSVQ (Zhou et al., 2024) further concentrate on constructing informative subtask representations to enable dynamic subtask allocation. However, although these explicit decomposition methods enhance behavioral diversity, they often introduce complex hierarchical structures or rely on latent-space inference, which decreases end-to-end training efficiency.

The second approach attributes the limited behavioral diversity to gradient conflicts among agents and mitigates this issue by partitioning the shared network into agent-specific subnetworks. Kaleidoscope (Li et al., 2024) and SNP (Kim & Sung, 2023) introduce learnable masks or structured pruning to carve out agent-specific parameter subsets within a shared backbone. More recently, GradPS (Qin et al.) directly uses inter-agent gradient conflicts to guide the construction of subnetworks and alleviates these conflicts by branching neurons identified as conflicting, thereby achieving state-of-the-art performance. However, the scalar conflict metrics used in this approach are not sufficiently expressive: reducing gradients to a single score inevitably entangles the directional and scale components of weight updates.

To fully characterize gradient conflicts among agents on shared parameters during training, we propose Geometric Gradient Decomposition analysis. Inspired by the neurobiological principle that separates feature selectivity (direction) (Hubel & Wiesel, 1962) from response intensity (scale) (Salinas & Abbott, 1996), we decompose gradients with respect to the weight vector into radial (scale) and tangential (direction) components. As seen in section 3, our empirical analysis uncovers a key insight: **agents largely agree on directional updates but substantially disagree on scale updates.** Consequently, while recent methods split the shared network into agent-specific subnetworks to mitigate conflicts, they also discard shared directional updates, thereby constraining training efficiency and final performance.

Guided by this Neuro-Geometric insight, we propose Hyperspherical Parameter Sharing (HPS). Instead of passively mitigating conflicts, HPS prevents them through architectural design. Specifically, HPS constrains the shared backbone weights onto a Riemannian manifold(unit hypersphere), thereby enforcing purely directional learning. Building on this, an agent-specific scale generator outputs multiplicative modulation factors to adjust each agent's activation scales, thus preserving heterogeneous response magnitudes without disrupting the shared directions. This structure allows agents to share a robust common representation while maintaining individual freedom in response scale, effectively resolving the optimization dilemma.

In summary, our main contributions are:

- We propose a Geometric Gradient Decomposition analysis inspired by neurobiology, showing that gradient conflicts in MARL arise primarily from scale rather than direction.

- We propose HPS, which combines Riemannian Backbone with agent-specific scale generator to structurally decouple weight direction from scale modulation.

- We evaluate HPS on SMAC, VMAS and Pedator Prey , demonstrating that it outperforms state-of-the-art baselines with limited parameter overhead.

## 2. Preliminaries

### 2.1. Decentralized Partially Observable Markov Decision Process

A general cooperative multi-agent task involving n agents can be formalized as a Decentralized Partially Observable Markov Decision Process (Dec-POMDP). A Dec-POMDP consists of a tuple $G = \langle I, S, A, P, R, \Omega, O, n, \gamma \rangle$, where: $I$ denotes the finite set of $n$ agents, $s \in S$ represents the true state within the global state space $S$, $A$ is the action space

of each agent's action $a_i$, which together form the joint action $a \in A^n$, $P(s'|s,a)$ is the state transition function determined by the environment, $R$ is the reward function providing common reward $r = R(s,a,s') \in \mathbb{R}$, $O$ is the observation function generating an individual observation from the observation space $\Omega$, i.e., $o_i \in \Omega$, and finally $\gamma \in [0,1)$ is a discount factor. The team's objective is to learn the policies that maximize the expected discounted accumulated reward $G_t \doteq \sum_{k=0}^{\infty} \gamma^k R_{t+k+1}$.

### 2.2. Gradient Conflict in Parameter Sharing

In the full Parameter Sharing paradigm, all agents share the same set of network parameters $\theta$. At this time, the joint optimization objective becomes the sum of all individual agent objectives, and the total gradient $g_{total}$ is the superposition of each agent's gradient $g_i$:

$$g_{total} = \nabla_\theta J(\theta) = \sum_{i=1}^{n} \nabla_\theta J_i(\theta) = \sum_{i=1}^{n} g_i. \quad (1)$$

Due to the varying requirements of heterogeneous tasks, the gradients $g_i$ generated by different agents often have inconsistent directions. Here, we refer to the classical definition of Gradient Conflict in PCGrad (Yu et al., 2020):

**Definition 2.1.** (Gradient Conflict). For any two agents $i$ and $j$, when the inner product of their gradient vectors is negative, i.e., $g_i \cdot g_j < 0$, we say that a gradient conflict exists between these two agents.

## 3. Geometric Gradient Decomposition Analysis

To alleviate the insufficiency in prior characterizations of gradient conflicts among agents on shared parameters during training, we propose Geometric Gradient Decomposition Analysis. Specially, we decompose gradients with respect to the weight vector into radial (scale) and tangential (direction) components and reveal the truth of scale conflicts hidden beneath conventional cosine similarity metrics.

### 3.1. Quantification of Parameter Vector Conflicts

**Radial-Tangential Decomposition:** Consider a weight vector $\mathbf{w} \in \mathbb{R}^d$, its evolution in the parameter space determines the network's behavior. We observe that the total gradient $\mathbf{g}$ can be uniquely decomposed into two orthogonal components relative to the current weight direction $\hat{\mathbf{w}} = \mathbf{w}/\|\mathbf{w}\|$:

$$\mathbf{g} = \underbrace{(\mathbf{g} \cdot \hat{\mathbf{w}})\hat{\mathbf{w}}}_{\mathbf{g}_\parallel \text{ (Radial/Scale)}} + \underbrace{(\mathbf{g} - (\mathbf{g} \cdot \hat{\mathbf{w}})\hat{\mathbf{w}})}_{\mathbf{g}_\perp \text{ (Tangential/Direction)}} . \quad (2)$$

The two components have distinct effects (proof in Appendix A.1): the radial term $\mathbf{g}_\parallel$ controls the weight-norm scale (gain/scale modulation), while the tangential term

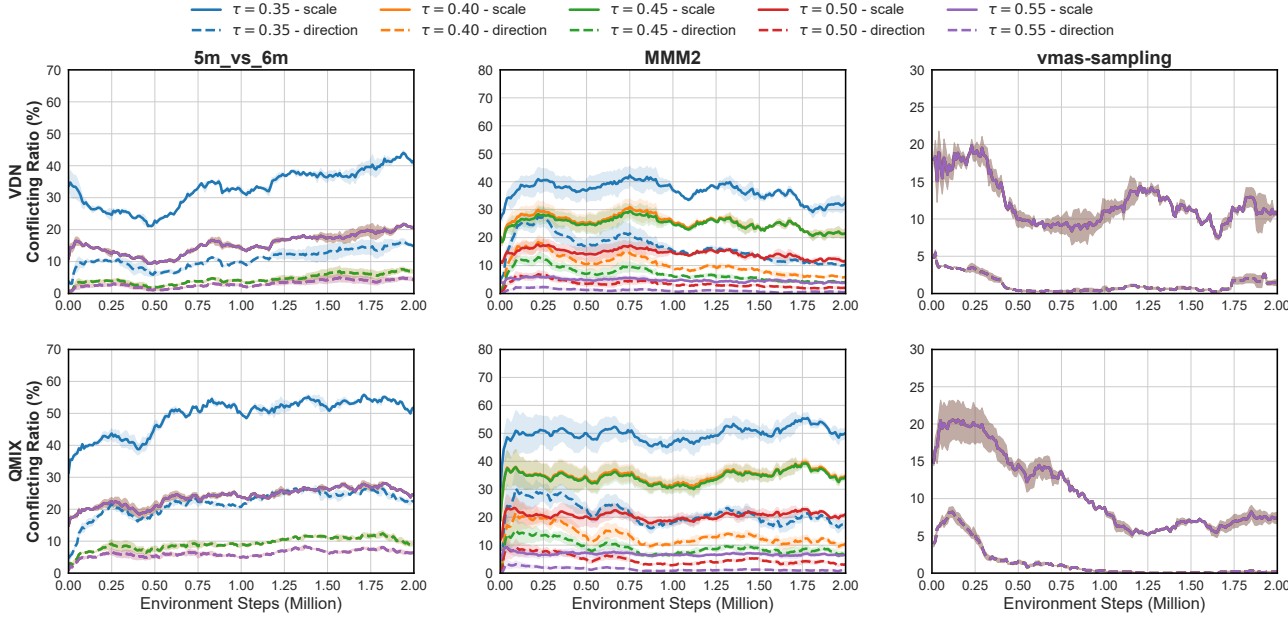

*Figure 1.* Geometric gradient decomposition analysis.

$g\perp$ rotates the weight direction (feature selectivity). This decomposition shows that conflicts in $g\|$ and $g\perp$ are fundamentally distinct.

While standard methods (e.g., PCGrad) rely on the overall cosine similarity, we argue that this scalar metric offers only a low-resolution view of optimization dynamics. A conflict in raw gradients is a sufficient but not necessary condition for component-wise disagreement (proof in Appendix A.2). Specifically, while any raw conflict indicates an underlying geometric divergence, the standard metric often fails to detect scale-specific conflicts that are overshadowed by direction agreement. By separating direction and scale via two geometrically decoupled metrics, our analysis yields a more comprehensive diagnosis:

**1. Scale Compatibility** ($S_{scale}$): This metric assesses the consistency of the radial components $g\|$. Since these components operate in a one-dimensional subspace, their similarity reduces to sign agreement:

$$S_{scale} = \text{sgn}(g_{i,\|} \cdot g_{j,\|}) \in +1, -1. \quad (3)$$

**2. Direction Consensus** ($S_{dir}$): This metric evaluates the alignment of the tangential components $g\perp$, reflecting whether agents agree on the direction of feature representation learning.

$$S_{dir} = \text{CosSim}(g_{i,\perp}, g_{j,\perp}). \quad (4)$$

**Conflict Criterion:** For either metric, a value of $S < 0$ signifies the occurrence of a specific type of gradient conflict (Scale or Direction).

Consider a layer consisting of $M$ parameter vectors, denoted as $\{\mathbf{w}_k\}_{k=1}^{M}$. At training step $t$, for each parameter vector $\mathbf{w}_k$, we collect the gradients backpropagated from all $n$ agents, forming the set $\mathcal{G}_{t,k} = \{g_{i,t,k}\}_{i=1}^{n}$ to measure the degree of conflict in the parameter vector by Vector-level Conflict Density.

**3. Vector-level Conflict Density** ($\rho_{k,t}(type)$): We first quantify the intensity of conflict specifically for the $k$-th parameter vector. The Conflict Density is defined as the proportion of agent pairs exhibiting negative geometric similarity on this specific vector:

$$\rho_{k,t}(type) = \frac{1}{N_{pairs}} \sum_{1 \leq i < j \leq n} \mathbb{I}(S_{type}(g_{i,t,k}, g_{j,t,k}) < 0). \quad (5)$$

where $type \in \{dir, scale\}$ and $N_{pairs} = \frac{n(n-1)}{2}$ is all possible pairs.

For a given threshold $\tau \in [0, 1]$, a parameter vector is flagged as a Conflicted Vector if its conflict density $\rho_{k,t}(type)$ exceeds $\tau$. This indicates that the parameter unit is subject to significant disagreement among agents:

$$C_{k,t}(type, \tau) = \mathbb{I}(\rho_{k,t}(type) > \tau). \quad (6)$$

**4. Layer-wise Conflict Ratio** ($\Phi_t(type, \tau)$): Finally, to quantify the prevalence of conflicts across the entire network

layer at time $t$, we calculate the proportion of vectors that are flagged as conflicted. This global metric, denoted as $\Phi_t(type, \tau)$, serves as the primary indicator for visualizing the optimization landscape:

$$\Phi_t(type, \tau) = \frac{1}{M} \sum_{k=1}^{M} C_{k,t}(type, \tau) \times 100\%. \quad (7)$$

where $M$ represents the total number of weight vectors in the layer. This metric effectively measures the extent of disagreement within the shared backbone.

### 3.2. Empirical Observations

To systematically study the distribution of gradient conflicts across learning paradigms and task settings, we expand our empirical evaluation. Specifically, we evaluate two canonical algorithms, VDN and QMIX, on three representative multi-agent scenarios: 5m_vs_6m (homogeneous, StarCraft II), MMM2 (heterogeneous, StarCraft II), and vmas-sampling (physics-based continuous control, VMAS). Each algorithm is evaluated with three random seeds for each figure to ensure result reliability.

We focus on the fully connected layer of the network and statistically analyzed the Layer-wise Conflict Ratio ($\Phi_t(type, \tau)$) during training process with different threshold $\tau = \{0.35, 0.40.0.45, 0.50.0.55\}$. Due to the limited number of agents in certain scenarios, the Layer-wise Conflict Ratio curves overlap across different thresholds in some plots.

As illustrated in Figure 1, we observe that in highly heterogeneous scenarios such as MMM2, where agent types differ substantially, all algorithms exhibit markedly higher conflicts in scale than in direction across thresholds. A similar pattern also appears in the homogeneous 5m_vs_6m setting, indicating that this phenomenon is not unique to heterogeneous scenarios. Even in low-conflict scenarios such as vmas-sampling, scale exhibits greater conflicts than direction. We argue that, to solve the task, the shared network must represent task-relevant features (weight direction), whereas each agent must adapt the response magnitude to its own needs (weight scale), leading to agreement in direction but conflicts in scale.

Our statistical results reveal a universal phenomenon across scenarios and algorithms:

- **High Direction Consensus:** The direction component of gradients ($g_\perp$) maintains high consistency among agents (low conflict ratio). This indicates that agents largely agree on what features to encode

- **Severe Scale Discrepancy:** Conversely, conflicts are predominantly concentrated in the radial component

($g_{||}$), indicating that agents largely disagree on response intensity.

## 4. Method

Building upon the Radial-Tangential Decomposition established in Section 3.1, we propose Hyperspherical Parameter Sharing. As illustrated in Figure 2, the framework decouple the policy into two orthogonal pathways: (1) **Riemannian Backbone (Direction Consensus):** Constrains the shared representation learning to the unit hypersphere manifold, forcing agents to agree on what features to extract ($g_\perp$). (2) **Agent-Specific Scale Generator (Scale Adaptation):** Re-introduces the necessary response intensity ($g_{||}$) via a agent-specific modulation mechanism.

### 4.1. Riemannian Backbone for Direction Consensus

To ensure the shared backbone learns universal features robust to heterogeneous scaling requirements, we constrain its weight space to the Riemannian manifold $\mathcal{S}^{d-1} = \{w \in \mathbb{R}^d : \|w\|_2 = 1\}$.

**Forward Pass (Hyperspherical Projection):** As seen in Figure 2(c), during the forward propagation, for an input observation $\mathbf{o}_i$ from agent $i$, the raw weights $\mathbf{v}_k$ are projected onto the manifold via $L_2$ normalization before computation:

$$\mathbf{h}_{dir}^{(i)} = \mathbf{W}_s \mathbf{o}_i, \quad \text{where } \mathbf{w}_k = \frac{\mathbf{v}_k}{\|\mathbf{v}_k\|}, \quad \forall k \in \{1, \ldots, d_{out}\}. \quad (8)$$

This operation structurally removes scale information from the backbone, ensuring that the feature extraction depends solely on the direction of the weights.

**Backward Pass (Geometric Gradient Filtering):** By applying the chain rule to the normalization step, the effective gradient update on the unnormalized weights $v$ becomes:

$$\frac{\partial \mathcal{L}}{\partial v} = \frac{1}{\|v\|} \underbrace{(I - ww^\top)}_{\mathbf{P}_\perp} \frac{\partial \mathcal{L}}{\partial w}. \quad (9)$$

Proof is in Appendix A.3. Here, $\mathbf{P}_\perp$ acts as a Geometric Filter. It projects the raw gradient onto the tangent space of the hypersphere, strictly enforcing orthogonality to the weight vector $w$. This provides a mathematical guarantee that gradient components attempting to alter the weight scale, identified as the main source of conflict in Section 3, are removed prior to updating the backbone.

### 4.2. Agent-Specific Scale Generator

Although the hyperspherical backbone guarantees stable directional sharing, heterogeneous tasks demand that agents apply varying degrees of scale. To recover this expressivity, we adopt an agent-specific scale generator inspired by biological Scale control (Salinas & Abbott, 1996). Specially,

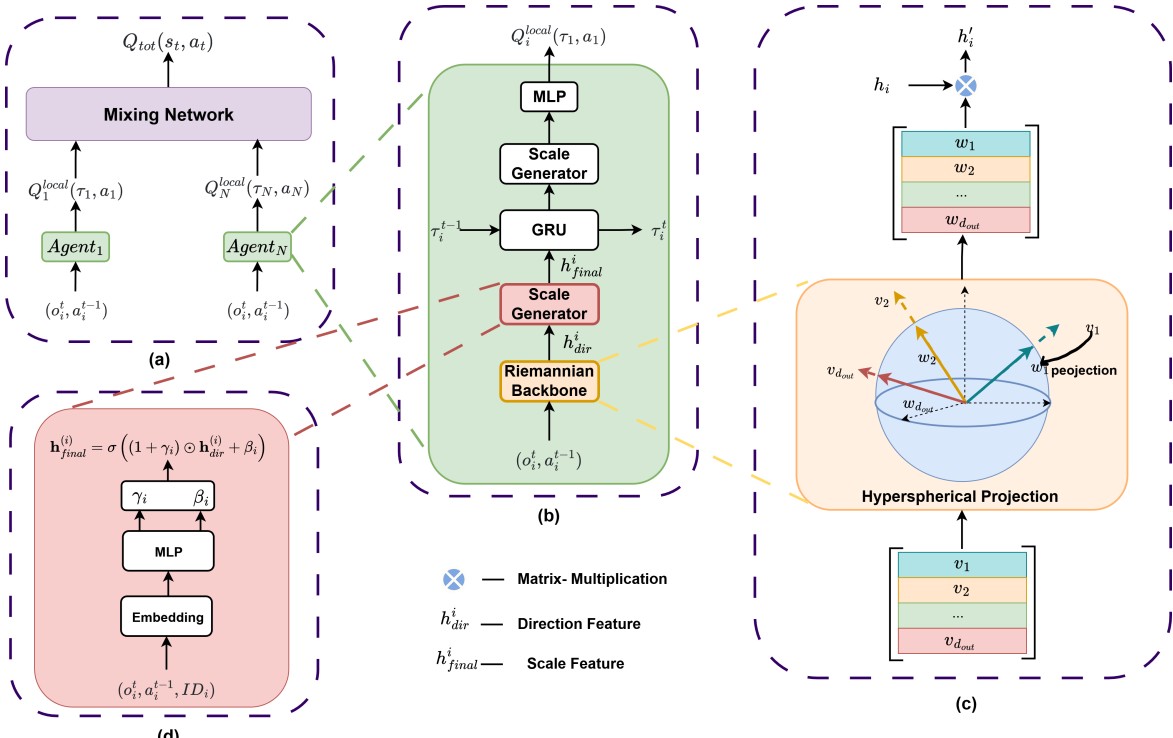

*Figure 2.* Overview of Hyperspherical Parameter Sharing (HPS). (a) The overall architecture. (b) The structure of shared individual Q-network. (c) The detail of Riemannian Backbone for Direction Consensus. (d) The detail of Agent-Specific Scale Generator.

we employ a generator network $\mathcal{G}_\phi$ to predict agent-specific scale parameters. Formally, to explicitly differentiate between distinct agents, we concatenate the agent identification $ID_i$ with the local observation $o_i$ to generate the scale ($\gamma_i$) and shift ($\beta_i$) vectors:

$$[\gamma_i, \beta_i] = \mathcal{G}_\phi(\mathbf{o}_i, \mathrm{ID}_i). \tag{10}$$

The final representation $h_{final}^{(i)}$ is synthesized via Feature-wise Linear Modulation:

$$\mathbf{h}_{final}^{(i)} = \sigma\left((1+\gamma_i)\odot\mathbf{h}_{dir}^{(i)}+\beta_i\right). \tag{11}$$

where $\odot$ denotes the Hadamard product and $\sigma$ is ReLU activation.

**Design Rationale:** This design decouples feature learning in the Riemannian Backbone from per-agent response scaling via the Agent-Specific Scale Generator. As $\mathcal{G}\phi$ is optimized separately, it can capture scale-related conflicts ($g\|$) while leaving the Riemannian Backbone unaffected.

### 4.3. Theoretical Guarantee: Geometric Isolation

We formally show how HPS resolves the optimization dilemma. Let the total aggregated gradient be $g_{total}$. As

derived in Eq. 2, this decomposes into radial ($g_\|$) and tangential ($g_\perp$) components.

Under HPS, the effective update $\Delta v$ on the shared backbone is governed by the projection operator $\mathbf{P}_\perp$ (from Eq. 9):

$$\Delta\mathbf{w} \propto \mathbf{P}_\perp\mathbf{g}_{total} = \mathbf{P}_\perp(\mathbf{g}_\| + \mathbf{g}_\perp) = \underbrace{\mathbf{P}_\perp\mathbf{g}_\|}_{=\mathbf{0}} + \underbrace{\mathbf{P}_\perp\mathbf{g}_\perp}_{=\mathbf{g}_\perp}. \tag{12}$$

Since the radial component $g_\|$ is by definition parallel to $w$, the projection $\mathbf{P}_\perp$ mathematically eliminates it ($\mathbf{P}_\perp g_\| = 0$). Consequently, the shared backbone is theoretically insulated from all scale-based conflicts. The optimization of response intensity is effectively re-routed to the Scale Generator $\mathcal{G}_\phi$, while the backbone $W_s$ evolves solely based on the consensus of direction ($g_\perp$). This achieves a provable structural disentanglement of shared commonalities and individual diversities. The detailed architecture of HPS can be found in the Appendix D.3.

## 5. Experiments

In this section, we conduct comprehensive experiments to evaluate the effectiveness of HPS. Our primary goals are to: 1)Assess the performance of HPS against other

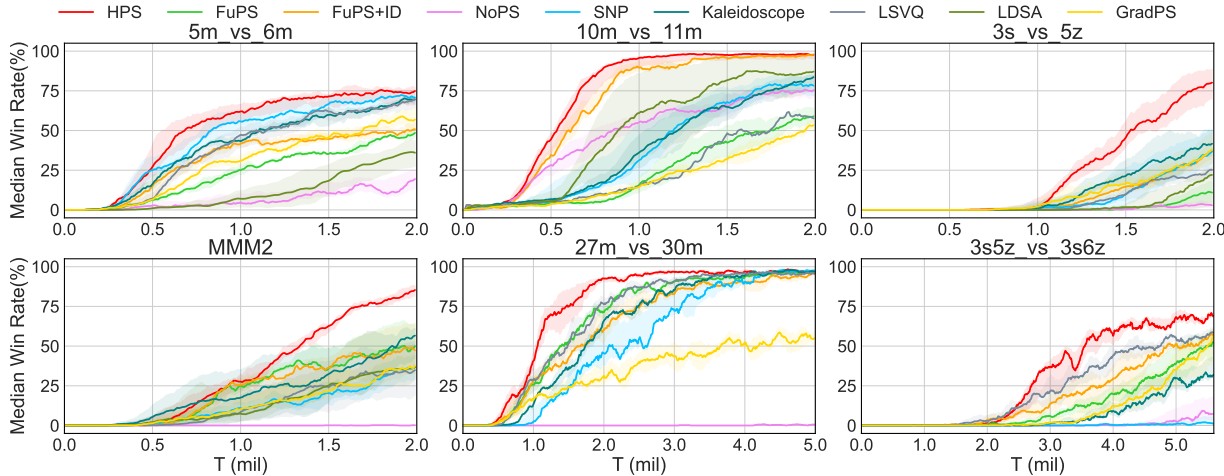

*Figure 3.* Performance comparison with baselines in different SMACv1 scenarios.

parameter sharing methods across diverse benchmarks. 2)Validate the robustness and generalization of HPS across different mixing networks. 3)Analyze the contribution of each component in HPS. 4)Validate the capability of HPS in resolving scale conflicts. 5) Analysis of Input Signals for Agent-Specific Scale Generator(Appendix E.2). 6) Analyze the role of $\beta$ in Agent-Specific Scale Generator(Appendix E.3). 7) Interpretability of Scale Coefficients(Appendix E.4). 8) Comparison of agent-network parameter usage(Appendix E.5). The source code for HPS is available in https://github.com/stimulus1594/HPS.

### 5.1. Experimental Setup

**Benchmarks.** We evaluate HPS across four diverse benchmarks spanning discrete and continuous domains. For discrete control, we employ **SMACv1** (Samvelyan et al., 2019), a StarCraft II-based micromanagement benchmark, alongside **SMACv2** (Ellis et al., 2023), which introduces randomized team compositions and limited fields of view. Complementing these, the **Vectorized Multi-Agent Simulator (VMAS)** (Bettini et al., 2022) serves as a continuous control testbed governed by physics dynamics (e.g., mass, velocity). Finally, we include a **Predator-Prey** (Hu et al., 2021) grid world, which explicitly tests cooperative incentives, serving as a classic testbed for resolving payoff-dominant coordination problems. More details are provided in Appendix C

**Baselines.** We compare HPS against a comprehensive suite of parameter sharing paradigms, including: (1) **FuPS+ID** and (2) **FuPS**, representing standard full parameter sharing with and without explicit agent indicators; (3)

**NoPS**, employing entirely independent policy networks; (4) **LDSA** (Yang et al., 2022) and (5) **LSVQ** (Zhou et al., 2024), which address heterogeneity through role modeling or subtask decomposition; (6) **Kaleidoscope** (Li et al., 2024), (7) **SNP** (Kim & Sung, 2023), and (8) **GradPS** (Qin et al.), representing state-of-the-art approaches for agent-specific subnetworks. We evaluate the performance of HPS and all baselines using the **QMIX** (Rashid et al., 2020b) value factorization framework to ensure fair comparison. Detailed implementations and hyperparameter configurations are available in Appendix D.

*Table 1.* Performance comparison with baselines in Predator-Prey.

| Methods | stag_hunt_s | stag_hunt_m | stag_hunt_l |
|---|---|---|---|
| FuPS | **70.08±0.87** | 114.58±1.57 | 174.48±0.38 |
| FuPS+ID | 69.62±0.96 | 115.39±0.54 | 172.91±2.41 |
| NoPS | 61.90±1.40 | 83.80±0.70 | 115.39±0.54 |
| SNP | 30.12±7.99 | 64.54±16.62 | 100.43±33.64 |
| Kalei | 56.00±12.41 | 104.20±6.39 | 75.53±10.32 |
| LSVQ | 69.95±0.90 | 115.57±1.82 | 170.85±0.54 |
| GradPS | 69.90±0.92 | 114.34±1.49 | 172.88±1.66 |
| HPS | 69.77±0.16 | **117.41±0.41** | **174.64±1.56** |

The overall effectiveness of the algorithm is illustrated in Figure 3, Figure 4, Figure 9(in Appendix E), Figure 5 and Table 1. Our primary observation is that HPS consistently achieves superior performance compared to state-of-the-art algorithms in most scenarios. This means that by decoupling scale and direction, HPS can effectively resolve conflicts while preserving parameter sharing. For Zerg series scenarios in Figure 4, we observe that the simplest parameter sharing baseline, FuPS, achieves highly robust performance,

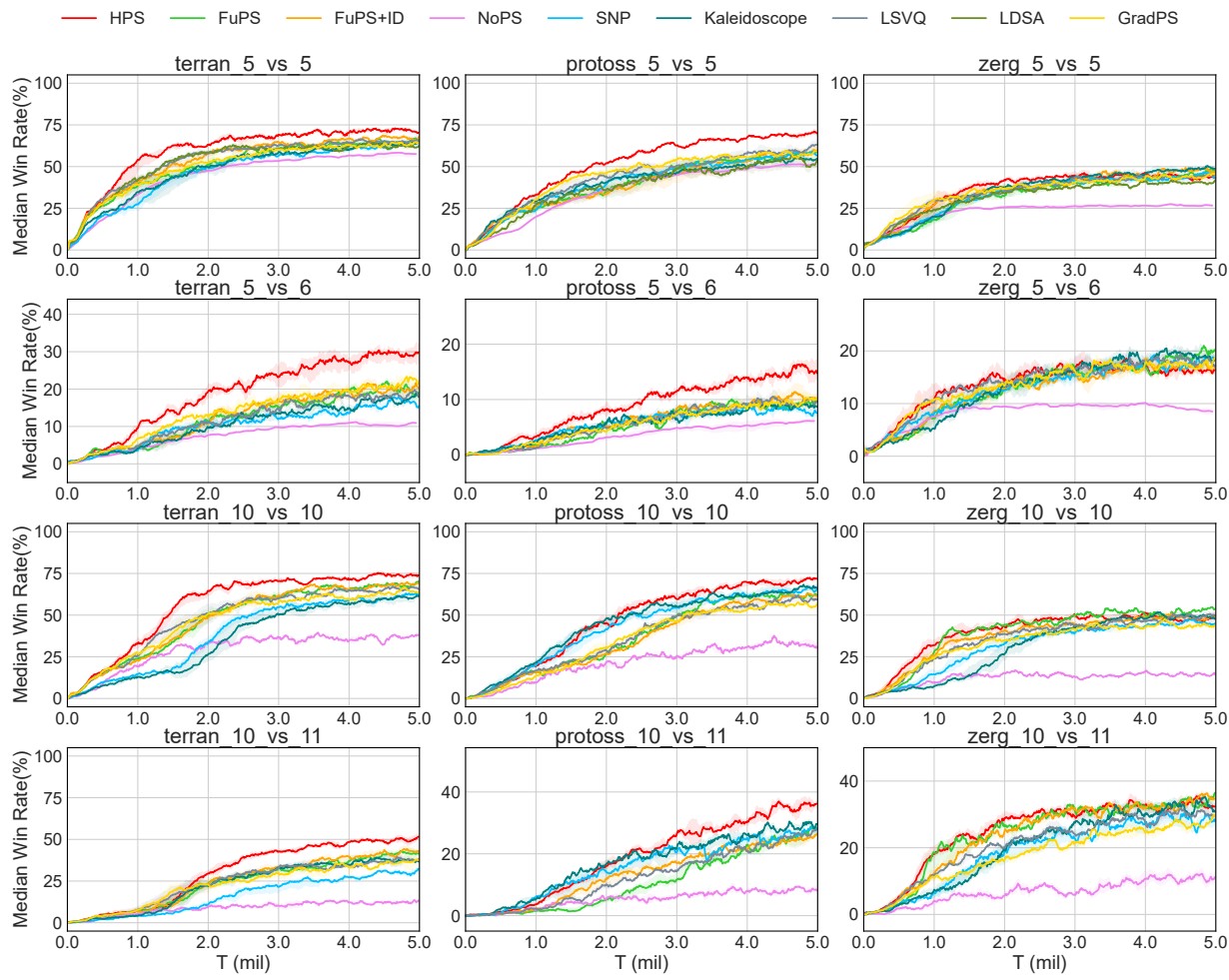

*Figure 4.* Performance comparison with baselines in different SMACv2 scenarios.

which indicates that strong parameter sharing is critical. However, as the number of agents increases, conflicts gradually emerge among agents, and the advantages of our method become progressively evident.

### 5.2. HPS Works for Multiple PS Agent Networks

To validate the robustness and generalization of our method across different mixing networks, we conduct experiments in the VMAS environment using various mixers, including VDN, QMIX, and QPLEX. As shown in the Figure 5, across diverse scenarios and value-mixing networks, HPS consistently improves performance, indicating that our geometric decoupling approach is fundamental and adaptable, effectively resolving conflicts and enhancing learning efficiency regardless of the specific mixing architecture employed.

### 5.3. Ablation Studies

To assess the contribution of each module, we conduct ablation studies on the Riemannian Backbone and the Agent-Specific Scale Generator separately. We remove the Riemannian Backbone module (HPS w/o Riemannian Backbone) and the Agent-Specific Scale Generator module (HPS w/o Scale Generator) from HPS, and conduct ablation studies on the homogeneous scenario $5m\_vs\_6m$, the heterogeneous scenario $MMM2$ in SMACv1, and the sampling scenario in VMAS. As shown in the figure 6, optimal performance is achieved only when the two modules work in coordination. Although removing either module leads to performance degradation, the results remain superior to QMIX, demonstrating the effectiveness of our method. More specifically, even without the Riemannian Backbone, our method still achieves excellent performance, further confirming that the

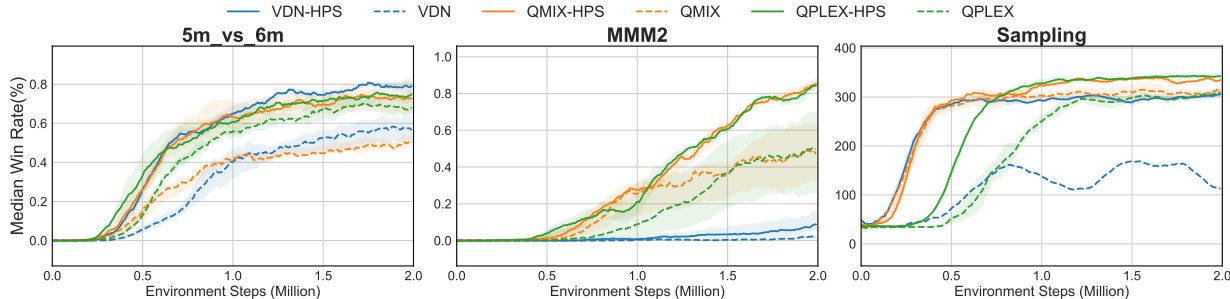

*Figure 5.* Performance comparison with different mixing network.

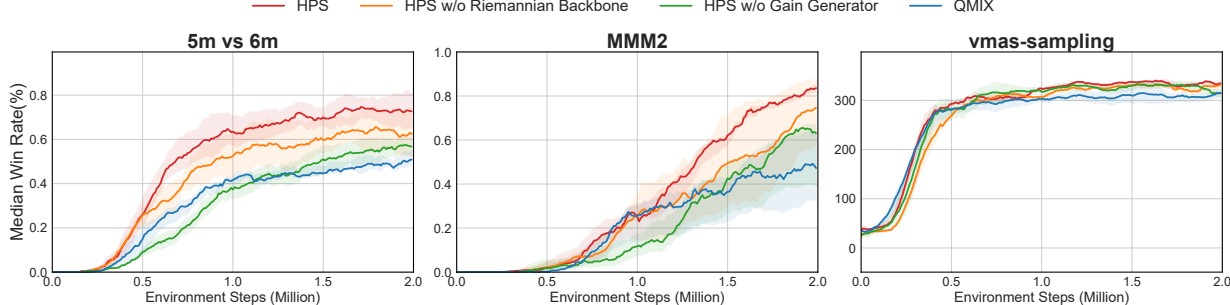

*Figure 6.* Performance comparison of modules ablation.

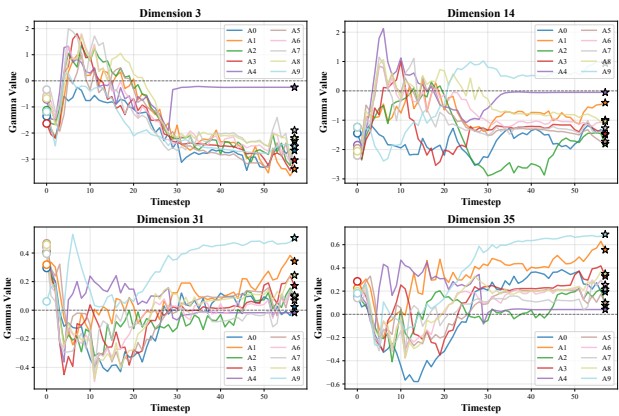

*Figure 7.* Scale visualization of MMM2.

current performance bottleneck mainly stems from scale conflicts. However, since the decoupling between direction and scale of gradient is not realized, scale conflicts can interfere with the learning of direction representations. When the Agent-Specific Scale Generator is removed, although performance drops substantially, it still surpasses the baseline

QMIX. This gain stems from our decoupling of direction and scale learning: despite persistent scale conflicts among agents, their gradients no longer interfere with direction learning, thereby enhancing the network's learning capacity.

### 5.4. HPS Effectively Mitigates Scale Conflicts

To validate the resolution of scale conflicts, we visualize the scale coefficients ($\gamma$) of three random backbone dimensions in the heterogeneous MMM2 map (Figure 7). First, a distinct divergence in scales across agents is observed, particularly in Dimensions 14 and 35. This confirms that HPS mitigates conflicts by allowing agents to maintain individualized response intensities. Second, the modulation is highly context-sensitive: Agent 9 dynamically shifts from suppressing to amplifying Dimension 14 as the episode progresses, reflecting adaptive feature selection. Finally, the generator exhibits an emergent auto-masking behavior, where scale coefficients for the died Agent 4 naturally vanish to zero, effectively filtering out noise from the aggregation.

# 6. Conclusion

We propose Geometric Gradient Decomposition Analysis and uncover a key insight: agents largely agree on directional updates but substantially disagree on scale updates. To address this, we propose HPS, which structurally decouples direction consensus and scale modulation. HPS consistently outperforms strong baselines across diverse benchmarks. However, in environments that require strong sharing, our method may slightly impair the degree of sharing and the algorithm's performance. In future work, we will further alleviate this issue and explicitly address conflicts in directional updates to further improve performance.

# Acknowledgements

This work was supported in part by the National Natural Science Foundation of China under Grant 62371201, the Foundation of National Key Laboratory of Multispectral Information Intelligent Processing Technology under grant 6142113-JCKY2022003, the National Natural Science Foundation of China under Grant Nos.624B2101, the Science and Technology Innovation Program of Hunan Province under Grant 2024RC1011, and the Major Basic Research Projects in Hunan Province under Grant 2026JC0001.

# Impact Statement

This work aims to advance the methodological foundations of multi-agent reinforcement learning by improving parameter-sharing efficiency under heterogeneous agent requirements. By decoupling shared directional representation learning from agent-specific scale modulation, HPS provides a general mechanism for enhancing coordination while preserving behavioral diversity. The proposed method may benefit cooperative multi-agent systems such as simulation-based coordination, swarm robotics, and autonomous control. As with other MARL methods, real-world deployment should carefully consider safety, robustness, and reliability, especially in high-stakes scenarios. Sufficient validation and human oversight are necessary to ensure that learned cooperative behaviors remain stable and aligned with intended objectives.

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

# A. Theoretical Derivations

## A.1. Radial-Tangential Gradient Decomposition

In this section, we provide mathematical proof for the geometric gradient decomposition presented in Section 3.1. We demonstrate that the radial component of the gradient strictly governs the scale dynamics, while the tangential component governs the direction dynamics.

### A.1.1. PRELIMINARIES

Let $\mathbf{w}(t) \in \mathbb{R}^d$ be a time-dependent weight vector evolving according to the continuous-time gradient flow dynamics:

$$\frac{d\mathbf{w}}{dt} = \dot{\mathbf{w}} = \mathbf{g} \tag{13}$$

where $\mathbf{g}$ is the total gradient. Let $\|\mathbf{w}\|$ denote the $L_2$ norm, and $\hat{\mathbf{w}} = \frac{\mathbf{w}}{\|\mathbf{w}\|}$ denote the unit direction vector.

Recalling Eq. 2 from the main text, the gradient $\mathbf{g}$ is decomposed as:

$$\mathbf{g} = \mathbf{g}_{\|} + \mathbf{g}_{\perp} \tag{14}$$

where $\mathbf{g}_{\|} = (\mathbf{g} \cdot \hat{\mathbf{w}})\hat{\mathbf{w}}$ is the radial component and $\mathbf{g}_{\perp} = \mathbf{g} - (\mathbf{g} \cdot \hat{\mathbf{w}})\hat{\mathbf{w}}$ is the tangential component.

### A.1.2. PROOF OF SCALE DYNAMICS

**Proposition A.1.** *The time evolution of the weight scale $\|\mathbf{w}\|$ is determined solely by the radial gradient component $\mathbf{g}_{\|}$.*

*Proof.* Let $r(t) = \|\mathbf{w}(t)\| = (\mathbf{w}^{\top}\mathbf{w})^{1/2}$. Differentiating with respect to time $t$:

$$\frac{d}{dt}\|\mathbf{w}\| = \frac{d}{dt}(\mathbf{w}^{\top}\mathbf{w})^{1/2} \tag{15}$$

$$= \frac{1}{2}(\mathbf{w}^{\top}\mathbf{w})^{-1/2} \cdot \frac{d}{dt}(\mathbf{w}^{\top}\mathbf{w}) \tag{16}$$

$$= \frac{1}{2\|\mathbf{w}\|}\left(\dot{\mathbf{w}}^{\top}\mathbf{w} + \mathbf{w}^{\top}\dot{\mathbf{w}}\right) \tag{17}$$

Since the dot product is symmetric and $\dot{\mathbf{w}} = \mathbf{g}$:

$$\frac{d}{dt}\|\mathbf{w}\| = \frac{1}{2\|\mathbf{w}\|}(2\mathbf{g}^{\top}\mathbf{w}) = \mathbf{g}^{\top}\frac{\mathbf{w}}{\|\mathbf{w}\|} = \mathbf{g} \cdot \hat{\mathbf{w}} \tag{18}$$

By definition, the scale of the radial component $\mathbf{g}_{\|}$ is $(\mathbf{g} \cdot \hat{\mathbf{w}})$. Furthermore, since $\mathbf{g}_{\perp}$ is orthogonal to $\hat{\mathbf{w}}$ (i.e., $\mathbf{g}_{\perp} \cdot \hat{\mathbf{w}} = 0$), we have:

$$\frac{d}{dt}\|\mathbf{w}\| = (\mathbf{g}_{\|} + \mathbf{g}_{\perp}) \cdot \hat{\mathbf{w}} = \mathbf{g}_{\|} \cdot \hat{\mathbf{w}} \tag{19}$$

Thus, the rate of change of the norm depends exclusively on the projection of the gradient onto the radial direction. $\qquad\square$

### A.1.3. PROOF OF DIRECTION DYNAMICS

**Proposition A.2.** *The time evolution of the weight direction $\hat{\mathbf{w}}$ is determined solely by the tangential gradient component $\mathbf{g}_{\perp}$.*

*Proof.* We differentiate the unit vector $\hat{\mathbf{w}} = \frac{\mathbf{w}}{\|\mathbf{w}\|}$ with respect to time using the quotient rule:

$$\frac{d}{dt}\hat{\mathbf{w}} = \frac{d}{dt}\left(\frac{\mathbf{w}}{\|\mathbf{w}\|}\right) = \frac{\dot{\mathbf{w}}\|\mathbf{w}\| - \mathbf{w}\frac{d}{dt}\|\mathbf{w}\|}{\|\mathbf{w}\|^2} \tag{20}$$

Substituting $\dot{\mathbf{w}} = \mathbf{g}$ and the result from Proposition 1 ($\frac{d}{dt}\|\mathbf{w}\| = \mathbf{g} \cdot \hat{\mathbf{w}}$):

$$\dot{\hat{\mathbf{w}}} = \frac{\mathbf{g}\|\mathbf{w}\| - \mathbf{w}(\mathbf{g} \cdot \hat{\mathbf{w}})}{\|\mathbf{w}\|^2} \tag{21}$$

$$= \frac{1}{\|\mathbf{w}\|}\left(\mathbf{g} - \frac{\mathbf{w}}{\|\mathbf{w}\|}(\mathbf{g} \cdot \hat{\mathbf{w}})\right) \tag{22}$$

$$= \frac{1}{\|\mathbf{w}\|}(\mathbf{g} - \hat{\mathbf{w}}(\mathbf{g} \cdot \hat{\mathbf{w}})) \tag{23}$$

Observing the term in the parentheses, we recognize the definition of the tangential component $\mathbf{g}_\perp = \mathbf{g} - (\mathbf{g} \cdot \hat{\mathbf{w}})\hat{\mathbf{w}}$. Therefore:

$$\dot{\hat{\mathbf{w}}} = \frac{1}{\|\mathbf{w}\|}\mathbf{g}_\perp \tag{24}$$

This confirms that the rotation of the weight vector (change in direction) is proportional to the tangential component of the gradient and inversely proportional to the current weight scale. $\square$

## A.2. Proof of Conflict Detectability

In this section, we provide the mathematical justification for the claim made in Section 3.1: *"A conflict in raw gradients is a sufficient but not necessary condition for component-wise disagreement."* We specifically demonstrate how standard scalar metrics can fail to detect scale-specific conflicts when direction agreement is strong.

### A.2.1. GEOMETRIC DECOMPOSITION OF THE INNER PRODUCT

Consider the gradients $\mathbf{g}_i$ and $\mathbf{g}_j$ from two agents $i$ and $j$. Based on the orthogonal decomposition established in Eq. 2, we have:

$$\mathbf{g}_k = \mathbf{g}_{k,\|} + \mathbf{g}_{k,\perp}, \quad \text{for } k \in \{i, j\} \tag{25}$$

where $\mathbf{g}_{k,\|}$ is parallel to the weight vector $\mathbf{w}$ and $\mathbf{g}_{k,\perp}$ is orthogonal to $\mathbf{w}$.

The standard measure of conflict, the inner product $\mathbf{g}_i \cdot \mathbf{g}_j$, can be expanded as:

$$\mathbf{g}_i \cdot \mathbf{g}_j = (\mathbf{g}_{i,\|} + \mathbf{g}_{i,\perp}) \cdot (\mathbf{g}_{j,\|} + \mathbf{g}_{j,\perp}) \tag{26}$$

$$= \underbrace{\mathbf{g}_{i,\|} \cdot \mathbf{g}_{j,\|}}_{\text{Scale Interaction}} + \underbrace{\mathbf{g}_{i,\|} \cdot \mathbf{g}_{j,\perp}}_{0} + \underbrace{\mathbf{g}_{i,\perp} \cdot \mathbf{g}_{j,\|}}_{0} + \underbrace{\mathbf{g}_{i,\perp} \cdot \mathbf{g}_{j,\perp}}_{\text{Direction Interaction}} \tag{27}$$

Since the radial and tangential components are mutually orthogonal, the cross terms vanish. Thus, the total inner product is simply the sum of the component-wise inner products:

$$\mathbf{g}_i \cdot \mathbf{g}_j = \mathbf{g}_{i,\|} \cdot \mathbf{g}_{j,\|} + \mathbf{g}_{i,\perp} \cdot \mathbf{g}_{j,\perp} \tag{28}$$

### A.2.2. SUFFICIENCY OF RAW CONFLICTS

**Proposition A.3.** *A conflict in the raw gradients ($\mathbf{g}_i \cdot \mathbf{g}_j < 0$) implies the existence of a conflict in at least one geometric component (Scale or Direction).*

*Proof.* Assume the premise $\mathbf{g}_i \cdot \mathbf{g}_j < 0$. We prove by contradiction. Suppose there is no component-wise conflict, i.e., there is consensus in both scale and direction:

$$\mathbf{g}_{i,\|} \cdot \mathbf{g}_{j,\|} \geq 0 \quad \text{and} \quad \mathbf{g}_{i,\perp} \cdot \mathbf{g}_{j,\perp} \geq 0 \tag{29}$$

Substituting this into Eq. (28), the sum of two non-negative terms must be non-negative:

$$\mathbf{g}_i \cdot \mathbf{g}_j = (\geq 0) + (\geq 0) \geq 0 \tag{30}$$

This contradicts the premise $\mathbf{g}_i \cdot \mathbf{g}_j < 0$. Therefore, a raw gradient conflict strictly necessitates that either $\mathbf{g}_{i,\|} \cdot \mathbf{g}_{j,\|} < 0$ (Scale Conflict) or $\mathbf{g}_{i,\perp} \cdot \mathbf{g}_{j,\perp} < 0$ (Direction Conflict). $\square$

A.2.3. NON-NECESSITY AND THE ILLUSION OF CONSENSUS

**Proposition A.4.** *The absence of a raw gradient conflict* ($\mathbf{g}_i \cdot \mathbf{g}_j > 0$) *does not imply the absence of component-wise conflicts. Specifically, a Scale Conflict can be overshadowed by strong Direction Consensus.*

*Proof.* Consider a case where agents disagree on scale but agree on direction:

1. **Scale Conflict:** Let $\mathbf{g}_{i,\|}$ and $\mathbf{g}_{j,\|}$ point in opposite directions along $\hat{\mathbf{w}}$. Let their dot product be $-C_{scale}$, where $C_{scale} > 0$.

2. **Direction Consensus:** Let $\mathbf{g}_{i,\perp}$ and $\mathbf{g}_{j,\perp}$ be aligned with a positive dot product $C_{dir} > 0$.

Substituting into Eq. (28):

$$\mathbf{g}_i \cdot \mathbf{g}_j = -C_{scale} + C_{dir} \tag{31}$$

If the directional agreement is sufficiently strong such that $C_{dir} > C_{scale}$, then:

$$\mathbf{g}_i \cdot \mathbf{g}_j > 0 \tag{32}$$

**Result:** The standard metric (Cosine Similarity or Inner Product) yields a positive value, indicating consensus. However, the underlying scale component exhibits a clear conflict. This example shows that raw gradient similarity may indicate overall consensus while still hiding scale-specific disagreement. $\square$

### A.3. Derivation of Geometric Gradient Filtering

In this section, we derive the gradient of the loss function $\mathcal{L}$ with respect to the unnormalized weights $\mathbf{v}$, given the normalization operation $\mathbf{w} = \frac{\mathbf{v}}{\|\mathbf{v}\|}$. This derivation formally justifies the *Geometric Gradient Filtering* mechanism described in Section 4.1.

A.3.1. FORWARD PASS DEFINITION

Let $\mathbf{v} \in \mathbb{R}^d$ be the raw weight vector. The normalized weight vector $\mathbf{w} \in \mathbb{R}^d$ is defined as:

$$\mathbf{w} = \frac{\mathbf{v}}{\|\mathbf{v}\|} \tag{33}$$

where $\|\mathbf{v}\| = (\mathbf{v}^\top \mathbf{v})^{1/2}$ is the $L_2$ norm. Note that by definition, $\|\mathbf{w}\| = 1$.

A.3.2. BACKWARD PASS DERIVATION

We seek to compute the gradient $\frac{\partial \mathcal{L}}{\partial \mathbf{v}}$ given the gradient from the subsequent layer $\frac{\partial \mathcal{L}}{\partial \mathbf{w}}$. By the chain rule:

$$\frac{\partial \mathcal{L}}{\partial \mathbf{v}} = \left( \frac{\partial \mathbf{w}}{\partial \mathbf{v}} \right)^\top \frac{\partial \mathcal{L}}{\partial \mathbf{w}} \tag{34}$$

where $\frac{\partial \mathbf{w}}{\partial \mathbf{v}}$ is the Jacobian matrix of $\mathbf{w}$ with respect to $\mathbf{v}$.

**Step 1: Compute the Jacobian $\mathbf{J} = \frac{\partial \mathbf{w}}{\partial \mathbf{v}}$** The $i$-th element of $\mathbf{w}$ is $w_i = v_i \|\mathbf{v}\|^{-1}$. The partial derivative with respect to the $j$-th element $v_j$ is:

$$\frac{\partial w_i}{\partial v_j} = \frac{\partial}{\partial v_j} (v_i \|\mathbf{v}\|^{-1}) \tag{35}$$

$$= \frac{\partial v_i}{\partial v_j} \|\mathbf{v}\|^{-1} + v_i \frac{\partial}{\partial v_j} (\|\mathbf{v}\|^{-1}) \tag{36}$$

Analyzing the first term: $\frac{\partial v_i}{\partial v_j} = \delta_{ij}$ (Kronecker delta). Analyzing the second term using the chain rule on $\|\mathbf{v}\| = (\sum v_k^2)^{1/2}$:

$$\frac{\partial}{\partial v_j} (\|\mathbf{v}\|^{-1}) = -\|\mathbf{v}\|^{-2} \frac{\partial \|\mathbf{v}\|}{\partial v_j} = -\|\mathbf{v}\|^{-2} \left( \frac{v_j}{\|\mathbf{v}\|} \right) = -\frac{v_j}{\|\mathbf{v}\|^3} \tag{37}$$

Substituting back:

$$\frac{\partial w_i}{\partial v_j} = \frac{\delta_{ij}}{\|\mathbf{v}\|} - \frac{v_i v_j}{\|\mathbf{v}\|^3} \tag{38}$$

In matrix notation, the Jacobian $\mathbf{J}$ is:

$$\frac{\partial \mathbf{w}}{\partial \mathbf{v}} = \frac{1}{\|\mathbf{v}\|}\mathbf{I} - \frac{1}{\|\mathbf{v}\|^3}\mathbf{v}\mathbf{v}^\top = \frac{1}{\|\mathbf{v}\|}\left(\mathbf{I} - \frac{\mathbf{v}\mathbf{v}^\top}{\|\mathbf{v}\|^2}\right) \tag{39}$$

**Step 2: Express in terms of normalized weights $\mathbf{w}$** Recall that $\mathbf{w} = \frac{\mathbf{v}}{\|\mathbf{v}\|}$. Thus, $\frac{\mathbf{v}\mathbf{v}^\top}{\|\mathbf{v}\|^2} = \left(\frac{\mathbf{v}}{\|\mathbf{v}\|}\right)\left(\frac{\mathbf{v}}{\|\mathbf{v}\|}\right)^\top = \mathbf{w}\mathbf{w}^\top$.
Substituting this into the Jacobian:

$$\frac{\partial \mathbf{w}}{\partial \mathbf{v}} = \frac{1}{\|\mathbf{v}\|}(\mathbf{I} - \mathbf{w}\mathbf{w}^\top) \tag{40}$$

**Step 3: Compute the final gradient** Now, multiply by the gradient $\mathbf{g} = \frac{\partial \mathcal{L}}{\partial \mathbf{w}}$:

$$\frac{\partial \mathcal{L}}{\partial \mathbf{v}} = \left[\frac{1}{\|\mathbf{v}\|}(\mathbf{I} - \mathbf{w}\mathbf{w}^\top)\right]\frac{\partial \mathcal{L}}{\partial \mathbf{w}} \tag{41}$$

$$= \frac{1}{\|\mathbf{v}\|}(\mathbf{I} - \mathbf{w}\mathbf{w}^\top)\frac{\partial \mathcal{L}}{\partial \mathbf{w}} \tag{42}$$

**Geometric Interpretation:** The matrix $\mathbf{P}_\perp = (\mathbf{I} - \mathbf{w}\mathbf{w}^\top)$ is the orthogonal projection matrix onto the tangent space of the hypersphere at $\mathbf{w}$. It explicitly removes the component of the gradient parallel to $\mathbf{w}$:

$$\mathbf{P}_\perp \mathbf{g} = \mathbf{g} - (\mathbf{w}^\top \mathbf{g})\mathbf{w} = \mathbf{g}_\perp \tag{43}$$

Thus, the update on the raw weights $\mathbf{v}$ is strictly proportional to the tangential component of the gradient, effectively filtering out any radial (scale-changing) signals.

## B. Related Works

### B.1. Value-based MARL

The Centralized Training with Decentralized Execution (CTDE) framework enhances agent collaboration, inspiring algorithms like VDN (Sunehag et al., 2017) and QMIX (Rashid et al., 2020b) which break down team value functions into agent-specific components by either simply summing the individual agent values or by mixing them via a continuous monotonic function, which limit their applicability in more general tasks. Weighted QMIX (Rashid et al., 2020a) further improves robustness by prioritizing beneficial actions. Additionally, PAC (Zhou et al., 2022) enhances value function representation in partially observable environments by incorporating counterfactual predictions. Moreover, to enable the value decomposition network to capture the structural information of multiple agents, some methods such as VAST (Phan et al., 2021), GOMARL (Zang et al., 2023) introduce agent grouping into the mixing network to perform hierarchical value mixing, thereby enhancing the expressive capability of the value decomposition network. However, these approaches enhance training efficiency through parameter sharing, but overlook the heterogeneity among agents.

### B.2. Parameter Sharing and Behavioral Diversity

While Parameter Sharing (PS) (Rashid et al., 2020b) serves as a cornerstone for sample efficiency in MARL, it neglects the varying decision requirements across agents, leading to insufficient behavioral diversity and degrade performance

**Explicit Role and Subtask Decomposition.** To address the limitations of full parameter sharing, one line of research focuses on explicit role modeling or subtask decomposition. Early works like ROMA (Wang et al., 2020a) and RODE (Wang et al., 2020b) introduce latent role concepts to decompose complex joint action spaces, guiding agents toward specialization. Building on this, LDSA (Yang et al., 2022) constructs informative subtask representations to enable dynamic subtask allocation. Similarly, LSVQ (Zhou et al., 2024) utilizes latent quantization to represent subtask variations, aiming to enhance coordination capabilities. However, while these decomposition methods improve behavioral diversity, they often introduce

complex hierarchical structures or rely on cumbersome latent-space inference, which can compromise end-to-end training efficiency and architectural generality.

**Agent-specific subnetwork.** The second approach attributes the limited behavioral diversity to gradient conflicts among agents and mitigates this issue by partitioning the shared network into agent-specific subnetworks. Methods such as SNP (Kim & Sung, 2023) and Kaleidoscope (Li et al., 2024) employ structured pruning and learnable masks, respectively, to isolate agent-specific parameter subsets within a shared backbone. Most recently, GradPS (Qin et al.) achieved state-of-the-art performance by identifying policy homogeneity through gradient conflicts and mitigating interference by physically branching conflicting neurons.

**Limitations of Prior Art.** Despite these advancements, existing agent-specific subnetwork methods, particularly GradPS, rely on scalar similarity metrics that conflate weight direction and scale updates. By treating all conflicts as destructive interference, these approaches may inadvertently discard valuable shared direction features, leading to network over-segregation and parameter inefficiency. Our work addresses this by geometrically decoupling these optimization objectives.

# C. Introduction for Environments

In this paper, we employed four experimental environments:SMACV1, SMACv2, VMAS, and Predator-Prey. Given the limited research on the recent introduction of some new environments, it is essential to provide a detailed overview of these environments. The following sections aim to familiarize readers with the objectives of these tasks and the corresponding evaluation metrics, offering a foundation for understanding this emerging field.

## C.1. SMACv1

*Table 2.* Descriptions of StarCraft II Challenge Scenarios

| Scenario | Ally Units | Enemy Units | Type | Level of Difficulty |
|---|---|---|---|---|
| 5m_vs_6m | 5 Marines | 6 Marines | Homogeneous Asymmetric | Hard |
| 10m_vs_11m | 10 Marines | 11 Marines | Homogeneous Asymmetric | Hard |
| 3s_vs_5z | 3 Stalkers | 5 Zealots | Homogeneous Asymmetric | Hard |
| MMM2 | 1 Medivac 2 Marauders 7 Marines | 1 Medivac 3 Marauders 8 Marines | Heterogeneous Asymmetric | Super Hard |
| 27m_vs_30m | 27 Marines | 30 Marines | Homogeneous Asymmetric | Super Hard |
| 3s5z_vs_3s6z | 3 Stalkers 5 Zealots | 3 Stalkers 6 Zealots | Heterogeneous Asymmetric | Super Hard |

SMAC (Samvelyan et al., 2019) is a simulation environment for research on cooperative multi-agent reinforcement learning (MARL), based on Blizzard's real-time strategy game StarCraft II. The environment provides various micromanagement combat scenarios and also allows users to customize scenarios to test different algorithms. In each scenario, the goal is to control different types of allied agents to move or attack in order to defeat enemy units. Enemy units are controlled by built-in heuristic AI, with difficulty adjustable from level 1 to 7. In our experiments, the game AI was set to the highest difficulty level (level 7). The version of StarCraft II used in the experiments is 2.4.10. Detailed scenario specifications are cataloged in Table 2.

## C.2. SMACv2

SMACv2 (Ellis et al., 2023) is developed to overcome the limited stochasticity present in the original SMAC environment. In SMACv2, the attack ranges vary across different unit types, and agents' fields of view are restricted to a sector shape rather than a full circle. At the beginning of each episode, team compositions and agent starting positions are randomly generated.

*Table 3.* Unit types used per-race in the SMACv2 scenarios.

| Race | Unit Types | Probability of Generation |
|---|---|---|
| Terran | Marine | 0.45 |
| | Marauder | 0.45 |
| | Medivac | 0.1 |
| Zerg | Zergling | 0.45 |
| | Baneling | 0.1 |
| | Hydralisk | 0.45 |
| Protoss | Stalker | 0.45 |
| | Zealot | 0.45 |
| | Colossus | 0.1 |

These changes make SMACv2 significantly more difficult. The version of StarCraft II used in the experiments is 2.4.10. This benchmark includes various scenarios categorized into Zerg, Terran, and Protoss, with each scenario allowing different number of agents to be configured . Here, we evaluate HPS and the baseline methods across all scenarios, with both sides having different number of agents: $zerg5x5$, $terran5x5$, $protoss5x5$, $zerg5x6$, $terran5x6$, $protoss5x6$, $zerg10x10$, $terran10x10$, $protoss10x10$, $zerg10x11$, $terran10x11$, $protoss10x11$. The agent types and their generation probabilities in the scenarios are shown in the Table 3.

## C.3. VMAS

To evaluate continuous control strategies, we utilize the Vectorized Multi-Agent Simulator (VMAS) (Bettini et al., 2022), a highly scalable framework designed for MARL that leverages a differentiable, vectorized 2D physics engine. VMAS provides a diverse suite of cooperative and competitive environments equipped with advanced physical interactions, such as elastic collisions, joints, and Lidar-based perception. For this study, we selected several representative scenarios configured to test collaborative efficiency and adaptability:

- **Balance:** Agents must cooperatively stabilize a spherical payload on a linear beam while transporting it to a target. This requires precise coordination to prevent the object from falling, which incurs significant penalties.

- **Sampling:** Three agents are spawned randomly in a workspace with an underlying gaussian density function composed of three gaussians modes. Agents need to collect samples by moving in this field.

- **Joint Passage Size:** Two robots of different sizes (blue circles),connected by a linkage through two revolute joints, need to cross a passage while keeping the linkage parallel to it and then match the desired goal position (green circles) on the other side.

## C.4. Predator-Prey

We evaluate HPS on the Stag Hunt game, a classic multi-agent grid world environment that epitomizes the conflict between **risk dominance** and **payoff dominance**. The environment consists of $N$ agents (predators) and an equal number of Stag and Hare entities inhabiting an $H \times W$ grid world.

**Task Dynamics:** The environment features two distinct prey types with varying incentives and coordination requirements:

- **Stag (High Payoff):** Yields a high collective reward ($r = +10$). However, capturing a stag is strictly cooperative, requiring at least **two agents** to simultaneously execute the *Catch* action within the capture range.

- **Hare (Low Risk):** Yields a lower reward ($r = +2$). A hare can be captured by a **single agent** executing the *Catch* action.

**The Coordination Challenge:** While there are no explicit heterogeneous traits, the agents face a profound coordination dilemma. A solitary hare represents a safe, immediate reward ("distraction"), whereas a stag offers a higher payoff but carries

the risk of coordination failure (if teammates do not join). In large-scale scenarios, varying local observations generate conflicting gradient signals: some agents may be biased towards nearby hares (high local response scale), potentially disrupting the team's direction consensus towards stags. This serves as a testbed for HPS's ability to filter out local distractions (via scale modulation) and maintain a unified cooperative direction.

**Configurations:** We conduct experiments across three difficulty levels to assess scalability. The detailed configurations are provided in Table 4.

*Table 4.* Detailed configurations for the Predator-Prey (Stag Hunt) scenarios.

| Config | Map Size | Agents | Stags | Hares | Episode Limit |
|--------|----------|--------|-------|-------|---------------|
| Small  | $20 \times 20$ | 6 | 6 | 6 | 150 |
| Medium | $25 \times 25$ | 10 | 10 | 10 | 150 |
| Large  | $30 \times 30$ | 15 | 15 | 15 | 150 |

## D. Implementation Details

### D.1. Introduction for baselines

We compare HPS against a comprehensive suite of parameter sharing paradigms, categorized into three groups:

**1) Standard Parameter Sharing Baselines:** We include (1) **NoPS** (No Parameter Sharing), a naive baseline where agents learn independent policies without weight sharing; (2) **FuPS** (Full Parameter Sharing), the standard approach sharing all parameters to maximize sample efficiency; and (3) **FuPS+ID**, which extends FuPS by appending one-hot agent identifiers to the input.

**2) Subtask and Role Decomposition Methods:** To evaluate performance against explicit heterogeneity modeling, we compare with (4) **LDSA** (Yang et al., 2022), which dynamically assigns learned subtasks to agents based on local observations; and (5) **LSVQ** (Zhou et al., 2024), which utilizes vector quantization (VQ) to construct discrete latent subtask representations for coordination.

**3) Parameter Isolation and Pruning Methods:** We also compare HPS against state-of-the-art methods that isolate parameters to resolve conflicts: (6) **SNP** (Kim & Sung, 2023), which employs structured pruning to isolate agent-specific sub-networks; (7) **Kaleidoscope** (Li et al., 2024), which uses learnable masks and matrix decomposition to capture shared and specific features; and (8) **GradPS** (Qin et al.), which dynamically branches futile neurons identified by conflicting gradient directions (negative cosine similarity).

### D.2. Algorithmic Description.

The pseudo-code of HPS is shown in Algorithm 1.

### D.3. Agent-Specific Scale Generator Details

The overall architecture of HPS is illustrated in Figure 8. Here, we detail the specific structural designs of the Agent-Specific Scale Generator and its integration with the shared backbone.

**Agent-Specific Scale Generator.** The generator consists of an encoding MLP, a GRU layer, and a decoding MLP.

- **Input:** At each time step $t$, the generator takes the concatenation of the local observation $o_i^t$, the last action $a_i^{t-1}$, and the agent's one-hot identity $ID_i$ as input.

- **Architecture:** The input is first processed by a fully connected (FC) layer, followed by a GRU to encode historical context, and finally projected by an output MLP to generate the scale coefficients $\gamma_i$.

- **Configuration:** To maintain a lightweight footprint, we set the hidden size of all internal layers (both MLPs and GRU) to 64.

- **Parameter Sharing:** Crucially, the parameters of the Scale Generator are shared across all layers and all agents. By conditioning on the unique Agent ID, a single generator instance can adaptively produce distinct scale signals for

---

**Algorithm 1** Hyperspherical Parameter Sharing (HPS)

---

**Require:** Batch size $B$, Learning rate $\eta$, Shared Backbone $\theta$, Scale Generator $\phi$
1: Initialize unnormalized backbone $\theta$ and scale generator $\phi$
2: **for** $e = 1$ to $M$ episodes **do**
3:     **while** Episode is not end **do**
4:         **for** each agent $i$ **do**
5:             *// 1. Context-Aware Modulation*
6:             Generate scales $\gamma_i, \beta_i = \mathcal{G}(o_i, a_i^{last}, ID_i; \phi)$
7:             *// 2. Riemannian Backbone Forward*
8:             Normalize backbone weights: $\mathbf{w}_k \leftarrow \mathbf{v}_k / \|\mathbf{v}_k\|$ for all $k$, forming $\mathbf{W}_s$
9:             Compute directional features: $\mathbf{h}_{dir}^{(i)} \leftarrow \mathbf{W}_s \mathbf{o}_i$
10:           Synthesize final representation: $\mathbf{h}_{final}^{(i)} \leftarrow \sigma\left((1 + \gamma_i) \odot \mathbf{h}_{dir}^{(i)} + \beta_i\right)$
11:           Compute $Q_i$ values and select action $a_i$ based on $\mathbf{h}_{final}^{(i)}$
12:         **end for**
13:         Execute joint action $\mathbf{a}$, store transition in $\mathcal{D}$
14:         *// 3. Optimization with Implicit Geometric Filtering*
15:         Sample batch $\mathcal{B} \sim \mathcal{D}$
16:         Compute TD target: $y_{s,a} \leftarrow r + \gamma \max_{a'} Q(s', a'; \theta^-, \phi^-)$
17:         Compute Loss: $\mathcal{L}(\theta, \phi) = (Q(s, a; \theta, \phi) - y_{s,a})^2$
18:         Update $\phi \leftarrow \phi - \eta \nabla_\phi \mathcal{L}$
19:         Update $\theta \leftarrow \theta - \eta \nabla_\theta \mathcal{L}$ {Gradients naturally projected via chain rule}
20:     **end while**
21:     Update target networks if required
22: **end for**

---

heterogeneous agents. This design ensures that the parameter overhead remains negligible and invariant to the number of agents, guaranteeing scalability.

**Hybrid Integration with Riemannian Backbone.** The main policy network (backbone) is composed of Riemannian Backbone and a recurrent layer (GRU) to handle partial observability.

- **Riemannian backbone:** For the Riemannian backbone, we adopt the proposed Hyperspherical Parameter Sharing, replacing standard weights with Riemannian weights on the unit hypersphere and introducing multiplicative scale factors to adjust each agent's scale.

- **GRU Layer:** Due to the complex gating mechanisms in the standard GRU architecture, directly imposing Riemannian constraints on its internal weights is non-trivial and may result in training instability. To address this, we adopt a pragmatic input modulation strategy: instead of modifying the GRU weights, we apply the generated scale coefficients to the *output* of the GRU layer. As shown in the ablation study, the key to resolving scale conflicts lies in the Agent-Specific Scale Generator; therefore, scale conflicts can be effectively mitigated even without Riemannian constraints.

### D.4. Implementation Details

All settings are identical to those provided in PyMARL2 (Hu et al., 2021). Results of the baselines we compare with are obtained using the publicly available code released by their authors. Each algorithm is evaluated using four independent training runs with different random seeds, and the resulting plots include the median performance shown in dark color as well as the $25\% - 75\%$ percentiles shown in the shaded area. We conducted our experiments on a platform with 2 Intel(R) Xeon(R) Platinum 8352 CPU 2.10GHz processors, each with 36 cores. Besides, we use a GeForce RTX 4090 GPU to facilitate the training procedure. The time of execution varies by scenario.

All methods are implemented and evaluated within the PyMARL2 framework (Hu et al., 2021), using **QMIX** (Rashid et al., 2020b) as the mixing network. To ensure a fair comparison, we follow the default PyMARL2 settings for all common

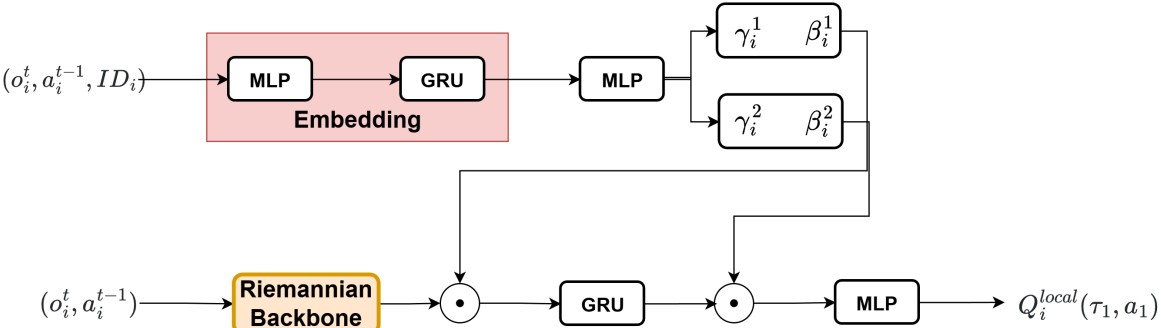

*Figure 8.* Details of Agent-Specific Scale Generator.

*Table 5.* Hyperparameter settings used in our experiments.

| Description | Value |
|---|---|
| Learning rate | 0.001 |
| Optimizer | Adam |
| Target network update interval | 200 |
| Gradient clipping global norm | 10 |
| Batch size | 128 |
| Replay buffer capacity | 5000 |
| Parallel rollout batch size | 8 |
| Discount factor | 0.99 |

training hyperparameters, while retaining the method-specific configurations reported in the corresponding papers. The detailed hyperparameter settings are summarized in Table 5.

## E. Additional Results

### E.1. Comparison on VMAS

The results are in Figure 9.

### E.2. Analysis of Input Signals for Agent-Specific Scale Generator

To rigorously evaluate the adaptive capability of the Agent-Specific Scale Generator, we examined its performance under different input configurations across *5m_vs_6m*, *MMM2*, and *vmas-sampling*: (1) **HPS (Obs+ID)**, (2) **HPS (Obs Only)**, and (3) **HPS (ID Only)**.

As shown in Figure 10, a consistent trend emerges: regardless of the input signal, **all HPS variants outperform the QMIX baseline**. This reinforces our core hypothesis that the geometric decoupling of direction and scale is the primary driver of performance, effectively mitigating conflicts even with partial information.

**Analysis of Input Modalities:**

- **HPS (ID Only):** While explicit IDs allow the generator to differentiate between agents, the resulting scale coefficients remain *static* throughout the episode. This lack of temporal flexibility prevents agents from adapting their roles to changing task phases, limiting peak performance.

- **HPS (Obs Only):** Without explicit IDs, the network must implicitly infer its identity from local observations, leading to *lower initial learning efficiency*. However, as the generator learns to disentangle identity from the observation space, performance accelerates, eventually approaching the optimal configuration.

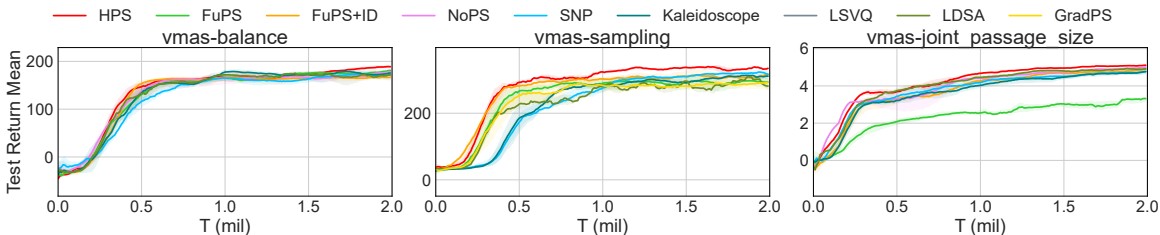

*Figure 9.* Performance comparison of vmas scenarios.

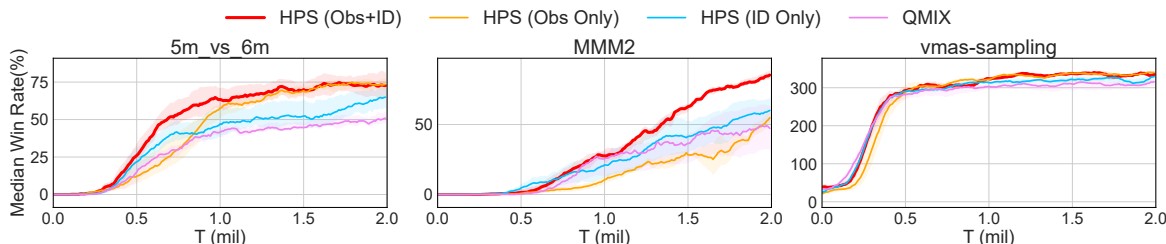

*Figure 10.* Performance comparison of different input for Agent-Specific Scale Generator.

- **HPS (Obs+ID):** This configuration yields the **optimal performance**. The explicit IDs provide immediate agent distinction, while the observation signals provide the necessary *contextual cues* for dynamic role adaptation. This synergy allows agents to adjust their response intensity in real-time according to the evolving task state.

### E.3. Analysis the role of $\beta$ in Agent-Specific Scale Generator

To further examine the role of the $\beta$ coefficient in the Agent-Specific Scale Generator, we evaluate a variant where all agents share a single $\beta$(HPS-Share-Beta). As shown in the Figure 11, HPS consistently achieves the best performance under all $\beta$ configurations, highlighting the importance of decoupling scale-related conflicts. The impact of $\beta$ on final performance depends on the scenario; for instance, under strongly heterogeneous settings such as MMM2, generating agent-specific $\beta$ via the Agent-Specific Scale Generator enables finer heterogeneity modulation and improves performance. In contrast, for 5m_vs_6m and VMAS-sampling, assigning an independent $\beta$ to each agent may weaken experience sharing among agents, resulting in performance degradation.

### E.4. Interpretability of Scale Coefficients

As illustrated in Figure 12, the scale coefficients exhibit significant variance across all dimensions, differentiating not only between distinct agents but also varying dynamically over time.

To examine the decision-making logic of the Agent-Specific Scale Generator, we extract scale-coefficient snapshots from four representative time points within a single *MMM2* episode. This fine-grained analysis reveals four interpretable characteristics:

1. **Tactical Temporal Sensitivity:** At $T = 5$ and $T = 15$, the engagement enters a complex confrontation phase requiring intensive micro-management. Correspondingly, we observe high-frequency fluctuations in the scale coefficients. This indicates that agents are actively modulating their feature attention to adjust positioning and select targets in response to rapidly changing battlefield dynamics.

2. **Functional Role Alignment:** We observe a clear contrast between Attackers and Medivac units. **Agents 0–8**

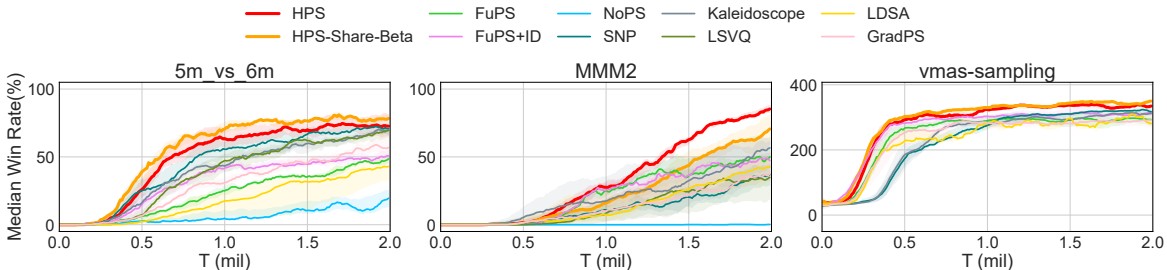

*Figure 11.* Performance analysis of role of $\beta$ in Agent-Specific Scale Generator.

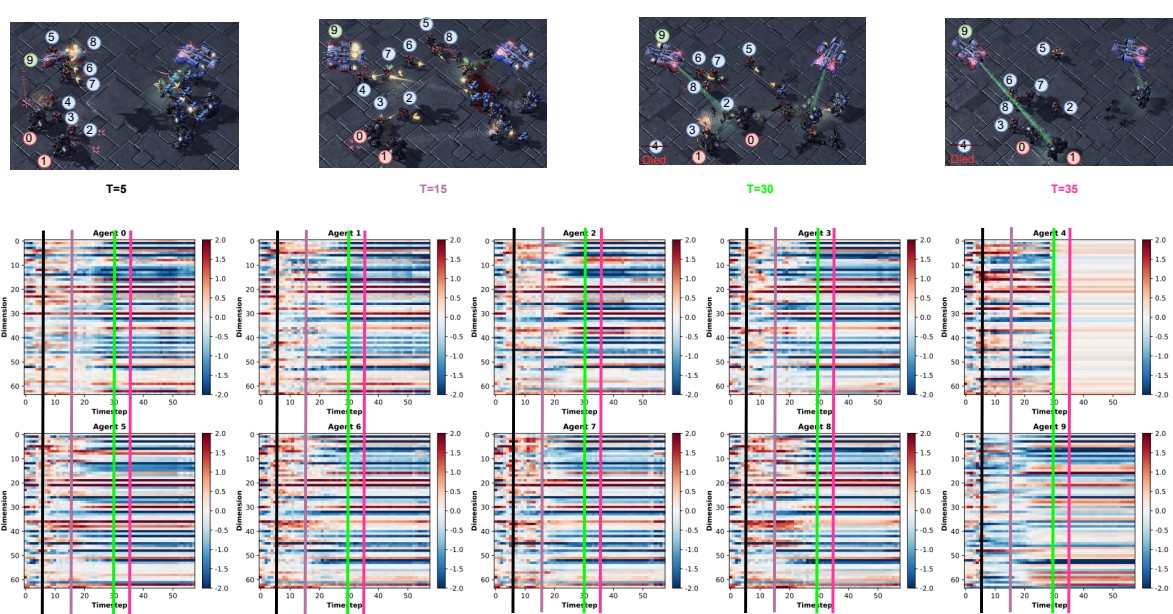

*Figure 12.* Temporal Evolution of Agent-Specific Scales.

**(Attackers)** exhibit scale patterns that emphasize enemy-status features, whereas **Agent 9 (Medivac)** shows the opposite pattern, focusing on ally-survival states. This suggests that the generator separates the distinct objectives of damage dealers and healers.

3. **Fine-Grained Sub-Group Differentiation:** Even within the attacker group, the model captures subtle unit-specific variations. While **Marauders (Agents 0–1)** and **Marines (Agents 2–8)** share broader trend similarities, they exhibit distinct intensity profiles in specific feature subspaces. Notably, across dimensions 50–60 during the intense combat window ($T = 5$ to $T = 15$), Marauders maintain consistently stronger scale compared to Marines, reflecting their role as heavy armored units with different tactical priorities.

4. **Robustness to Attrition (Auto-Masking):** Post $T = 30$, following the death of Agent 4, its scale coefficients across all dimensions collapse to a near-zero distribution. This reconfirms the generator's capability to automatically mask out irrelevant signals from died agents, preserving the integrity of the collective representation.

*Table 6.* **Parameter Efficiency Analysis.** We report the number of agent-network parameters relative to the Full Parameter Sharing (FuPS) baseline (normalized to 100%). HPS introduces a fixed overhead from the scale generator, yet scales favorably in large scenarios (e.g., 27m_vs_30m): its parameter usage (233%) is substantially lower than branching-based methods such as GradPS (286%) and LDSA (533%).

| SCENARIO | FuPS | HPS | FuPS+ID | SNP | GRADPS | KALEIDOSCOPE | LDSA | LSVQ | NoPS |
|---|---|---|---|---|---|---|---|---|---|
| 5M_VS_6M | 100% | **254%** | 101% | 129% | 161% | 264% | 394% | 317% | 505% |
| 10M_VS_11M | 100% | **248%** | 102% | 126% | 200% | 283% | 434% | 423% | 1019% |
| MMM2 | 100% | **242%** | 102% | 123% | 235% | 297% | 417% | 415% | 1017% |
| 27M_VS_30M | 100% | **233%** | 104% | 121% | 286% | 326% | 533% | 714% | 2797% |

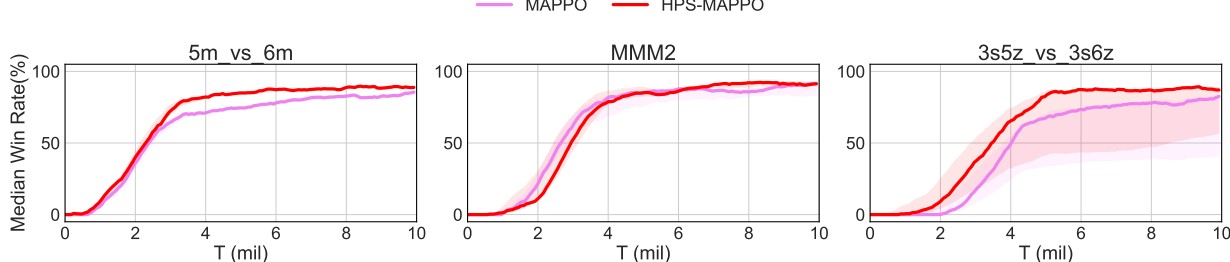

*Figure 13.* Performance comparison between MAPPO-based HPS and the MAPPO baseline.

### E.5. Usage of Agent Network Parameters.

Table 6 compares agent-network parameter usage across scenarios, normalized to FuPS. HPS augments the shared backbone with an Agent-Specific Scale Generator, which introduces a relatively larger overhead in small scenarios (e.g., $5m\_vs\_6m$: 254% vs. GradPS 161%). As the scenario scale increases, the parameter advantage of HPS becomes increasingly evident. In particular, in $27m\_vs\_30m$, GradPS rises to 286% and LDSA reaches 533%, whereas HPS remains stable and even drops to 233% in relative terms, since the generator incurs a largely fixed cost and captures heterogeneity via modulation rather than capacity growth. Overall, HPS is substantially more parameter-efficient than explicit role/subtask modeling (LDSA, 394–533%) and fully independent policies (NoPS, $> 1000\%$), suggesting that neuro-geometric modulation offers a compact alternative to parameter isolation or role decomposition.

### E.6. HPS Based on MAPPO

To examine whether HPS generalizes beyond value-decomposition methods, we further apply the proposed architecture to MAPPO (Yu et al., 2022). We implement HPS based on the official codebase and keep the same training hyperparameters as the MAPPO baseline for a fair comparison. As shown in Figure 13, results suggest that HPS is not tied to a specific value-decomposition framework, but can also enhance policy-based MARL methods through geometric decoupling.

### E.7. Additional Comparative Experiments

We further incorporate ACORM (Hu et al., 2024) and CDS (Li et al., 2021) as additional competitive baselines. All methods are evaluated with the same QMIX mixing network. Except for the necessary PyMARL2 specific settings (Hu et al., 2021), as summarized in Table 5, for consistent implementation, we strictly follow the officially released code and original hyperparameters of all baselines. As seen in Figure 14, HPS still maintains competitive or superior performance against these strong baselines.

### E.8. More Results of Gradient Conflict Pattern in GRU Layers

In the main text, we analyze the gradient conflict pattern on the fully connected layers and observe that scale-related conflicts are consistently stronger than direction-related conflicts. To further examine whether this phenomenon also exists

*Figure 14.* Additional Comparative Experiments.

in recurrent modules, we extend the same geometric gradient decomposition analysis to the GRU layers of the agent network. Specifically, we analyze both input-to-hidden and hidden-to-hidden weight matrices in the GRU, covering the new, reset, and update gates. The analysis is conducted on representative SMAC scenarios, including $10m\_vs\_11m$, $5m\_vs\_6m$, and $MMM2$. As shown in Figure 15, the same qualitative pattern can be observed across different GRU components and scenarios: scale-related conflicts are consistently larger than direction-related conflicts.

These results indicate that the scale-dominant conflict pattern is not limited to simple fully connected layers, but also appears in recurrent structures that encode temporal dependencies. This further supports our core observation that agents tend to agree on shared feature directions while requiring different response magnitudes. Therefore, the proposed HPS architecture is well motivated not only for feed-forward representation learning, but also for recurrent agent networks commonly used in MARL.

## F. Limitations and Future Work

HPS consistently outperforms strong baselines across diverse benchmarks. However, in environments that require strong sharing, our method may slightly impair the degree of sharing and the algorithm's performance. In future work, we will further alleviate this issue and explicitly address conflicts in directional updates to further improve performance.

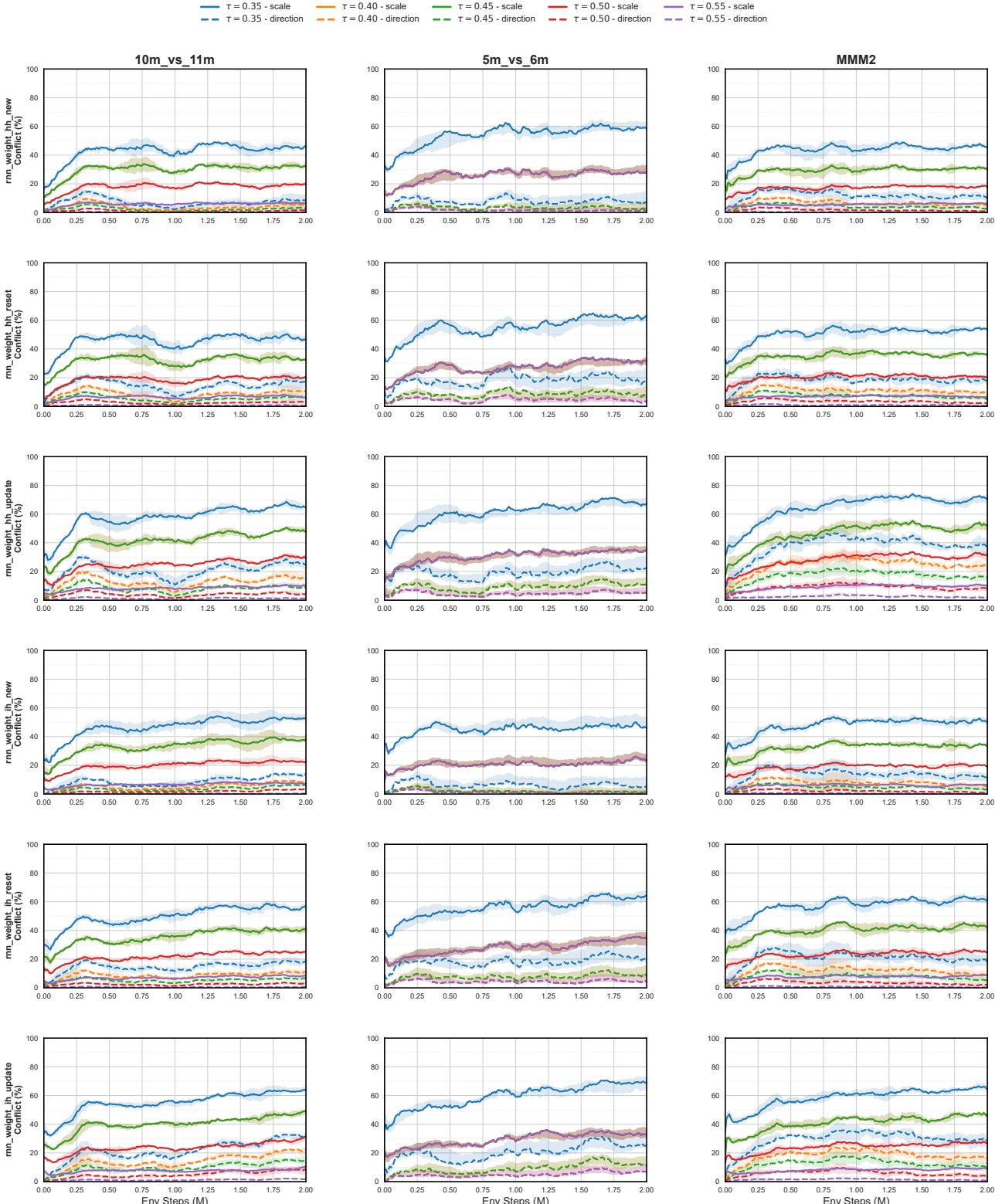

*Figure 15.* Gradient conflict pattern in GRU layers.

