# OpenReview forum: "HPS: Hyperspherical Parameter Sharing for Efficient Multi-Agent Reinforcement Learning"
_ICML.cc/2026/Conference — ICML 2026 regular_

### Official Review · Reviewer_6bdc · 2026-03-08

**Soundness:** 2
**Presentation:** 3
**Significance:** 3
**Originality:** 3
**Overall Recommendation:** 4
**Confidence:** 4

**Summary:**

This paper studies parameter sharing in cooperative MARL and argues that gradient conflicts among agents are primarily caused by disagreements in the scale component of updates rather than the direction component. To support this, the paper introduces a geometric decomposition of gradients into radial and tangential parts, and proposes Hyperspherical Parameter Sharing (HPS), which retains only directional learning within a shared hyperspherical backbone, and an agent-specific scale generator to model heterogeneous response magnitudes. The method is evaluated on SMACv1, SMACv2, VMAS, and Predator-Prey, with comparisons against several parameter-sharing and heterogeneity-aware baselines.

**Compliance With Llm Reviewing Policy:**

Affirmed.

**Final Justification:**

The rebuttal fully addresses my concerns.

**Key Questions For Authors:**

see weaknesses.

**Limitations:**

yes

**Strengths And Weaknesses:**

**Strengths**

1.This paper identify two important components of gradient in MARL, which is interesting and could advance the heterogeneous MARL.

2.The Geometric Gradient Decomposition Analysis part is quite good and easy-to-understand.

3.The experimental results show impressive performance improvement compared to baselines.

4.The scale visualization is helpful.


**Weaknesses**

W1.The motivation of Riemannian Backbone for Direction Consensus is confusing to me. The results in Sec. 3 show that agents maintain high direction consensus (Line 211), and yet the motivation of Riemannian Backbone is to enhance direction consensus (Line 171). If direction consensus is observed to be high enough, why is it important to further enhance it?

W2.Following W1, how high does the direction consensus need to be? Furthermore, what do higher $g_{\|}$ and $g_{\perp}$ indicate? Is there a correlation between the overall performance and $g_{\|}$ and $g_{\perp}$? I suggest the authors to report the changes of $g_{\|}$ and $g_{\perp}$ before and after adapting the proposed method.

W3.HPS encode the agent ID into the policy network. Does the baselines also do this? It seems agent ID is important for non-parameter-sharing SMAC.

---

> ### Author Rebuttal · Authors · 2026-03-31
>
> We sincerely thank for the constructive review and the recognition of our gradient decomposition analysis, impressive performance improvement, and helpful scale visualization.
>
> ## W1: Motivation of Riemannian Backbone
>
> We think our original wording was misleading. The Riemannian Backbone is not introduced to further enhance direction consensus. Rather, its role is to preserve the already aligned directional signal by structurally removing radial/scale interference from shared updates, thereby supplying the Scale Generator with a **cleaner direction-related feature** and making the learning of agent-specific scaling easier.
>
> The core logic is as follows. First, our analysis in Sec. 3 shows that agents are often already **relatively aligned in update direction**, while suffering much stronger **scale conflicts**. Second, by constraining the shared weights on the unit hypersphere, the Riemannian Backbone **eliminates the radial component of the gradient**, so the shared backbone only receives directional updates. This is beneficial because even under high directional agreement, conflicting magnitudes can still distort the aggregated shared update.
>
> This design is also important at the **framework level**. Without the Riemannian Backbone, the Scale Generator would need to handle not only agent-specific scale modulation, but also residual scale conflicts left in the shared backbone, which makes the learning problem significantly harder. Our ablation in Sec. 5.4 (Figure 6) supports this: **HPS w/o Riemannian Backbone** consistently underperforms full HPS across all tested scenarios, despite still using the Scale Generator. We will revise Sec. 4.1 accordingly and clarify that the backbone is introduced for **clean direction-scale decoupling**, rather than for increasing direction consensus itself.
> ## W2: Threshold for direction consensus? Correlation between metrics and performance? Changes before/after HPS?
>
> To our knowledge, there is **no universal threshold** for when direction consensus is “high enough”; this is task-dependent (see also our response to Reviewer Dd7Z Q1). Nevertheless, across the **tested settings**, we consistently observe that **scale conflict dominates direction conflict**. This is the main motivation behind HPS: rather than further improving direction consensus itself, HPS is designed to primarily address the **scale-related interference**. In our notation, higher $g_{\parallel}$ indicates a larger **scale-related conflict**, while higher $g_{\perp}$ reflects stronger disagreement in the orthogonal component.
>
> After introducing HPS, scale is explicitly routed to the **Scale Generator**, so the original full-network scale-conflict metric is no longer directly comparable before and after adaptation. We therefore measure scale conflict using the **gradient cosine similarity of the entire Scale Generator**. On both **5m_vs_6m** and **MMM2**, this quantity is **close to zero**, far below the corresponding baseline levels in **Figure 1**. This is consistent with the intended mechanism of HPS: scale-related variation is largely isolated into a dedicated branch, while the shared backbone receives cleaner directional signals. We will add more scenarios in the revision.
>
> ## W3: Does HPS encode agent ID? Do baselines also use agent ID?
>
> HPS encodes agent ID into the Scale Generator. The shared Riemannian Backbone does **not** receive agent ID.
>
> **Except for FuPS, all other baselines also use Agent ID:**
>
> | Method | Agent ID? | How |
> |--------|:-:|------|
> | FuPS | No | Fully shared |
> | FuPS+ID | Yes | ID concat to observation |
> | NoPS/SNP/Kaleidoscope/LSVQ/LDSA| Yes | ID as input; method-specific mechanisms |
> | **HPS** | **Yes** | **ID input to Scale Generator only** |
>
> Since all methods except FuPS use Agent ID, the comparison is fair.
>
> We further added input ablations (**Obs+ID / Obs / ID**). Results show that HPS does **not** rely on ID alone: on **5m_vs_6m**, **Obs Only** is comparable to full HPS (**73.54 vs. 72.66**), and on **vmas-sampling** it is slightly stronger (**340.44 vs. 334.74**), while **ID Only** is consistently worse. On the more heterogeneous **MMM2** task, however, full **Obs+ID** is clearly best (**85.35** vs. **54.79/60.06**). Thus, ID is helpful but not sufficient, and the main gain comes from the proposed **direction-scale decoupling**, not simply from encoding ID.
>
> | Scenario       | Method         |  Result      |
> |----------------|----------------|--------------|
> | 5m_vs_6m       | HPS (Obs+ID)   | 72.66 ± 9.22 |
> |        | HPS (Obs) | 73.54 ± 5.96 |
> |        | HPS (ID)  | 64.94 ± 10.79 |
> |        | QMIX           | 50.88 ± 5.83 |
> | MMM2           | HPS (Obs+ID)   | 85.35 ± 3.87 |
> |            | HPS (Obs) | 54.79 ± 9.28 |
> |            | HPS (ID)  | 60.06 ± 16.70 |
> |            | QMIX           | 47.27 ± 25.28 |
> | vmas-sampling  | HPS (Obs+ID)   | 334.74 ± 4.05 |
> |   | HPS (Obs) | 340.44 ± 2.49 |
> |   | HPS (ID)  | 328.91 ± 16.18 |
> |   | QMIX           | 315.34 ± 13.61 |

---

> > ### Author Rebuttal · Reviewer_6bdc · 2026-04-01
> >
> > Thanks for the rebuttal. I have still have some concerns.
> >
> > Q1: *"On both 5m_vs_6m and MMM2, this quantity is close to zero..."* Does this description refer to Fig. 7? If so,  why is the two figures (1 and 7) with two different y-coordinates comparable ?
> >
> > Q2: Can the authors provide the comparison of the scales or scale conflict ratios between HPS and baselines? (This is my original intent of the last question in W2)

---

> > > ### Author Response · Authors · 2026-04-02
> > >
> > > Thank you very much for this thoughtful follow-up. We sincerely apologize for the confusion caused by our previous wording. The question is highly insightful and exceptionally helpful.
> > >
> > > **Regarding Q1**. They are **not**. Specifically, **Figures 1 and 7 measure different quantities**, and therefore their y-axes should not be interpreted on the same scale.
> > >
> > > **Regarding Q2**.
> > > More concretely, as seen in https://anonymous.4open.science/r/data-3F89/rebuttal.png, in the original network, scale is represented by a shared learnable vector $\gamma$. In that setting, for each scale parameter $\gamma_i$, if the proportion of gradient conflicts across agents exceeds a threshold $\tau$, we identify this parameter as conflicting. Figure 1 therefore reports the proportion of conflicting parameters under different thresholds:
> > > $
> > > \mathrm{Dim}(\gamma_{\text{conflict}}) / \mathrm{Dim}(\gamma).
> > > $
> > >
> > > However, under HPS, scale is no longer a directly optimized shared parameter vector. Instead, $\gamma$ is generated by an additional scale generator network. As a result, the optimization gradients that would originally act directly on $\gamma$ are absorbed into the generator parameters. In this sense, the original scale-conflict metric is **not directly comparable** before and after adaptation.
> > >
> > > This is exactly why the reviewer’s question is so valuable: it reveals a subtle but crucial issue in how scale conflict should be measured after reparameterization. We sincerely thank the reviewer for pointing this out.
> > >
> > > To enable a more appropriate before-and-after comparison for HPS, we instead adopt a **parameter-level optimization-conflict metric** in the spirit of PCGrad[1], and evaluate  whether there exists an optimization conflict at $\gamma_i$ between each pair of agents with respect to parameters of the scale generator network. Based on this, we further compute the proportion of conflicting parameters under different thresholds $\tau$.
> > >
> > > Following the reviewer’s original intent, we further provide the comparison of scale-related conflict between HPS and the baseline. The results are shown below.
> > >
> > > ### Proportion of conflicting parameters (scale only)
> > >
> > > |                  |           0.35 |           0.40 |           0.45 |           0.50 |           0.55 |
> > > |------------------|---------------:|---------------:|---------------:|---------------:|---------------:|
> > > | MMM2-HPS         | 18.72% ± 3.64% | 12.50% ± 2.50% |  9.28% ± 2.46% |  4.66% ± 1.64% |  1.26% ± 0.52% |
> > > | MMM2-QMIX        | 50.12% ± 3.40% | 34.84% ± 1.41% | 34.38% ± 1.17% | 21.05% ± 1.05% |  6.52% ± 0.12% |
> > > | 5m\_vs\_6m-HPS   | 20.31% ± 3.81% | 13.16% ± 2.32% | 13.16% ± 2.32% |  7.95% ± 1.10% |  7.95% ± 1.10% |
> > > | 5m\_vs\_6m-QMIX  | 51.84% ± 1.68% | 24.65% ± 0.74% | 24.65% ± 0.74% | 24.65% ± 0.74% | 24.65% ± 0.74% |
> > >
> > > These results support our claim that, after the HPS reparameterization, optimization conflict in the scale-related component becomes dramatically smaller than in the original shared-scale formulation.
> > >
> > > Again, we sincerely thank the reviewer for this exceptionally insightful and technically sharp question. It helped us identify an ambiguity in our original presentation and significantly improved the clarity of our analysis. In the revision, we will update the corresponding text and figures.
> > >
> > > [1] Gradient Surgery for Multi-Task Learning

---

### Official Review · Reviewer_ZUUb · 2026-03-09

**Soundness:** 2
**Presentation:** 3
**Significance:** 2
**Originality:** 3
**Overall Recommendation:** 3
**Confidence:** 3

**Summary:**

This paper studies gradient conflict in parameter-sharing multi-agent reinforcement learning. Its central claim is that the main source of conflict under parameter sharing comes not from directional disagreement between gradients, but from differences in gradient scale. To support this claim, the paper introduces a geometric analysis framework that decomposes gradients into radial and tangential components, and reports that scale conflict is consistently stronger than direction conflict across several settings. Building on this observation, the paper proposes HPS, which combines a hyperspherical shared backbone for learning direction information with an agent-specific scale generator implemented through feature-wise modulation to recover individual differences. The method is evaluated on multiple benchmarks, where it is shown to achieve promising performance.

**Compliance With Llm Reviewing Policy:**

Affirmed.

**Final Justification:**

Concerns about the experimental setting still remain. In the current results, both FuPS and FuPS+ID perform well, which suggests that the present experimental design may still be insufficient to fully reveal the performance limitations caused by parameter sharing. Therefore, the current results are still not strong enough to convincingly support the conclusion that HPS can effectively mitigate the performance limitations introduced by parameter sharing.

**Key Questions For Authors:**

1. The experimental results are not yet fully convincing. In particular, SMACv1 is a standard benchmark on which many role-based or task-decomposition methods have already shown strong performance. However, in Figure 3, several baselines appear unexpectedly weak, especially on maps such as MMM2 and 3s5z_vs_3s6z, which raises concerns about the fairness of the comparison. In addition, Table 1 shows that FuPS and FuPS+ID outperform the other baselines by a clear margin, but this phenomenon is not sufficiently discussed. A similar issue appears in Figure 4: aside from HPS, most methods show very limited differences on most maps. I would encourage the authors to increase the number of random seeds and include more competitive baselines, such as ACORM[1] and CDS[2], to ensure a more thorough and fair comparison.

2. Could the authors provide gradient conflict decomposition results on more tasks and at more network layers, in order to better support the claim that scale conflict is the dominant source of conflict beyond the current experimental settings?

[1] Attention-guided Contrastive Role Representations for Multi-Agent Reinforcement Learning. ICLR 2024.
[2] Celebrating Diversity in Shared Multi-Agent Reinforcement Learning. NeurIPS 2021.

**Limitations:**

yes

**Strengths And Weaknesses:**

Strengths

1. The method is clearly designed: the hyperspherical backbone and the scale generator play distinct roles, and the overall framework is conceptually coherent.

2. The experimental section covers multiple benchmarks and compares against several representative baselines, making the evaluation reasonably broad.

3. Clear paper writting, the paper is will organized and easy to follow.

Weaknesses

1. The theoretical part mainly shows that hyperspherical constraints eliminate the radial component of the gradient. This is more a geometric property than a rigorous theoretical justification for improved performance, optimization stability, or generalization. As a result, the current theory only provides limited support for the claimed empirical advantages.

2. The paper’s main conclusion that conflict mainly arises from scale rather than direction currently reads more like an empirical observation than a well-established general principle. The evidence provided is still insufficient to support such a strong and broadly applicable claim.

3. The current experiments do not fully rule out an alternative explanation: namely, that the gains may come primarily from the additional scale generator and feature modulation capacity, rather than from the proposed direction-scale decoupling itself.

---

> ### Author Rebuttal · Authors · 2026-03-31
>
> We sincerely thank for the careful reading and the recognition of our clearly designed method with conceptually coherent framework, broad experimental coverage, and clear paper writing.
>
> ## W1: The theoretical part
>
> We agree that the theory is a mechanistic explanation of HPS: hyperspherical constraints remove the radial component from shared updates and restrict learning to the tangential direction. We will clarify this scope in the revision.
>
> ## W2: Scale-vs-direction conflict as an empirical trend
>
> We will revise the paper to present the scale-vs-direction result as a **stable empirical trend in the tested settings**, not a universal principle. Added analyses across multiple scenarios, seeds, and **GRU layers** (Reviewer Dd7Z Q1,Reviewer un9m Q2)show the same qualitative pattern: **scale conflict consistently exceeds direction conflict**, although the gap varies by scenario. We also find that HPS mainly reduces **scale conflict** while largely preserving directional agreement. Thus, we present this as a repeated empirical finding, supported by the geometric analysis, rather than a general theorem.
>
>
> ## W3: Gains may come primarily from the Scale Generator, not from direction-scale decoupling
>
> We added **QMIX-Large** and **FiLM-QMIX (QMIX + agent-specific scale)**. **QMIX-Large** does not improve over QMIX, showing that extra capacity alone is insufficient. **FiLM-QMIX** captures part of benefit, but remains far below HPS on **MMM2**. This suggests that the main gain comes from the proposed **direction-scale decoupling**, not from extra capacity or generic modulation alone.
>
> | Scenario | Method | Result|
> |-------|-------|----------|
> |5m_vs_6m|HPS|72.66±9.22|
> ||QMIX-Large|47.66±12.20|
> ||FILM-QMIX|68.78±15.59|
> |MMM2|HPS|85.35±3.87|
> ||QMIX-Large|34.80±32.13|
> ||FILM-QMIX|53.03±18.54|
> ## Q1: Fairness concerns — baselines appear weak on MMM2 and 3s5z_vs_3s6z
>
> **SMACv1.** All baselines are run in **PyMARL2** with **SMAC 2.4.10** under identical configurations, so the comparison within our study is controlled and fair. Our **FuPS+ID (QMIX)** results are also consistent with published results under comparable settings [3,5]. We note that **absolute scores** on hard heterogeneous maps such as **MMM2** and **3s5z_vs_3s6z** can be sensitive to the **PyMARL/SMAC version**, which is also consistent with prior reports [1,2,4].
>
> Following the reviewer’s suggestion, we added two stronger baselines, **ACORM** and **CDS**. The conclusion remains unchanged: **HPS is still the best method** across all tested settings, including **5m_vs_6m** (**72.66** vs. **53.12/42.97**), **MMM2** (**85.35** vs. **77.13/0.00**), **3s5z_vs_3s6z** (**73.83** vs. **62.70/48.44**), and **vmas-sampling** (**334.74** vs. **306.49/314.74**).
>
> Except for the necessary **PyMARL2-specific settings**, we use the **official released code** and **original hyperparameters** of all baselines. We therefore believe the remaining differences mainly stem from the **PyMARL2 / benchmark version**, and we are adding **more random seeds** to further verify this.
>
> | Scenario| Method | Result|
> |---------------|---|----------|
> | 5m_vs_6m|HPS|**72.66±9.22**|
> ||QMIX|50.88±5.83|
> ||ACORM|53.12±15.77|
> ||CDS|42.97±11.89|
> |MMM2|HPS|**85.35±3.87**|
> ||QMIX|47.27±25.28|
> ||ACORM|77.13±26.11|
> ||CDS|0.00±30.95|
> |3s5z_vs_3s6z|HPS|**73.83±2.72**|
> ||QMIX|60.94±3.16|
> ||ACORM|62.70±4.54|
> ||CDS|48.44±10.08|
>
>
>
> **Table 1.**
> We also provide additional gradient conflict analysis on **stag_hunt** scenarios. The results show that, while **scale conflict (parallel) is consistently larger than direction conflict (perpendicular)** across all thresholds, the **overall level of conflict is relatively low** in these tasks. This likely explains why simpler fully shared methods such as **FuPS / FuPS+ID** already perform well in such scenarios.
>
> | Scenario | Threshold |Scale|Direction|
> |----------------|-----------|-----------|-----------|
> | stag_hunt_m | th35| 8.24± 0.27| 1.29± 0.90|
> || th40| 5.31± 0.16| 0.78± 0.62|
> || th45| 5.31± 0.16| 0.55± 0.47|
> || th50 | 3.20± 0.16| 0.23 ± 0.16|
> || th55| 0.82± 0.04| 0.00± 0.00|
>
> **SMACv2.** Our results are also consistent with prior reports such as **Kaleidoscope** [6] under matched hyperparameters. The relatively small gaps among methods mainly reflect the inherent difficulty of **SMACv2**. Despite this, **HPS still achieves consistently better performance**, which supports our claim that **scale conflict** is an important bottleneck in parameter sharing.
>
> References.
> [1]Gradient-Protected Value Decomposition for Cooperative Multi-Agent Reinforcement,AAAI 2026.
> [2]Autonomous Partner Selection for Cooperative Multi-Agent Reinforcement Learning,AAAI 2026.
> [3]Learning Disentangled Multi-Agent World Model for Decentralized Control,ICLR 2026.
> [4]Automatic Grouping for Efficient Cooperative Multi-Agent Reinforcement Learningg,NeurIPS 2023.
> [5]MA2E,ICLR 2025.
> [6]Kaleidoscope,NeurIPS2024.
>
>
> ## Q2: Analysis on more tasks and layers
>
> See W2 above.

---

> > ### Author Rebuttal · Reviewer_ZUUb · 2026-04-04
> >
> > Thanks for the reply and the extensive additional experiments. These results have addressed some of my concerns, but I still have doubts about the fairness of the experimental comparisons. First, although differences in experimental settings across different versions may affect the results to some extent, I am still not fully convinced that the current comparisons are fair. Second, some observations in Table 1 and Figure 4 have not been sufficiently explained. Specifically, in Table 1, it is unclear why FUPS performs clearly better than other parameter-sharing methods. The authors still need to clarify whether this suggests that the current experimental design is not rigorous enough, or that the current environment cannot reflect the advantages of partial parameter sharing or other methods designed to reduce the limitations of parameter sharing. In Figure 4, the fact that FUPS either outperforms the other baselines or shows almost the same performance as them is also not sufficiently explained, which weakens the support for the related conclusions. For these reasons, I have decided to maintain my current score.

---

> > > ### Author Response · Authors · 2026-04-06
> > >
> > > We thank the reviewer for the detailed feedback and for raising these important concerns.
> > >
> > > ## Q1: Fairness concerns
> > > To further address fairness, we additionally evaluate HPS under **PyMARL1**, a widely used framework. The results are consistent with PyMARL2, confirming that our conclusions are **not tied to a specific implementation or version**.
> > >
> > > We compare against strong parameter-sharing baselines (**ACORM, CDS, GradPS**) under an identical **QMIX** setup with **4 seeds**. As shown in the table, **HPS achieves competitive or best performance across all scenarios**, consistent with our PyMARL2 findings.
> > >
> > > | Scenario |ACORM| CDS       |GradPS| **HPS** |
> > > |---|-----------|-----------|--------------|---|
> > > |5m_vs_6m|0.679±0.042| 0.597±0.058| 0.652 ± 0.059 | **0.656 ± 0.070** |
> > > |10m_vs_11m|0.819±0.055| 0.818±0.011 |0.821 ± 0.029 | **0.888 ± 0.021** |
> > > |3s_vs_5z|0.620±0.034|0.513± 0.089| 0.851 ± 0.114 | **0.918 ± 0.074** |
> > > |3s5z_vs_3s6z|0.513±0.227|0.627±0.157| 0.006 ± 0.008 | **0.760 ± 0.231** |
> > >
> > > We will include these results in the revised paper.
> > >
> > > ## Q2: Why does FUPS outperform other parameter-sharing methods in Table 1?
> > >
> > > We further analyze this on **stag_hunt** scenarios. Although **scale conflict dominates direction conflict**, the **overall conflict is very low**, with **direction conflict nearly zero**.
> > >
> > > This implies agents already share consistent **directional representations**, and interference mainly comes from **scale differences**, allowing simple fully shared methods (e.g., **FuPS / FuPS+ID**) to perform well.
> > >
> > > In contrast, partial sharing methods that **separate both scale and direction** may unnecessarily break useful directional sharing, reducing efficiency. This aligns with our design: **HPS preserves directional sharing while resolving scale conflict**.
> > >
> > > | Scenario | Threshold |Scale|Direction|
> > > |----------------|-----------|-----------|-----------|
> > > | stag_hunt_m | th35| 8.24± 0.27| 1.29± 0.90|
> > > || th40| 5.31± 0.16| 0.78± 0.62|
> > > || th45| 5.31± 0.16| 0.55± 0.47|
> > > || th50 | 3.20± 0.16| 0.23 ± 0.16|
> > > || th55| 0.82± 0.04| 0.00± 0.00|
> > >
> > > ## Q3： Performance in SMACV2
> > > We thank the reviewer for this observation. While **FUPS performs strongly in some SMACv2 scenarios**, this does not contradict our claims.
> > >
> > > First, SMACv2 introduces higher variability, and **agent IDs no longer carry stable semantics**, forcing methods to rely on observation-based generalization rather than ID shortcuts.
> > >
> > > Second, consistent with other scenarios, **scale conflict clearly dominates direction conflict**, indicating that inter-agent interference mainly arises from **scale differences**.
> > >
> > > Thus, the results further support our design: **HPS targets the dominant scale conflict while preserving shared directional learning**. We will revise the paper to better clarify this point.
> > >
> > > | Scenario| Threshold | Scale| Direction|
> > > |---------------|---|---|---|
> > > | protoss_5_vs_6|th35|43.20%±2.11%|17.93%±0.98%|
> > > ||th40|20.70%±0.55%|6.91%±0.35%|
> > > ||th45|20.70%±0.55%|6.91%±0.35%|
> > > ||th50|20.70%±0.55%|3.40%±0.04%|
> > > ||th55|20.70%±0.55%|3.40%±0.04%|
> > >
> > >
> > > ## Q4: Does the experimental design reflect the advantages of partial parameter sharing?
> > >
> > > We thank the reviewer for this question.
> > >
> > > We evaluate across diverse MARL benchmarks with varying heterogeneity and difficulty, including **stag_hunt**, **SMACv1**, **SMACv2**, and **VMAS**, covering both low-conflict and high-heterogeneity regimes.
> > >
> > > To better reflect partial-sharing advantages, we further extend experiments to **more VMAS scenarios**(specifically designed for heterogeneous multi-agent scenarios).
> > >
> > > | Scenario (steps) | HPS | QMIX | GradPS|LDSA| Kalei   |SNP|
> > > |---|---|---|---------|---------|---------|---|
> > > | balance| **184.39 ± 3.31** |164.64 ± 5.58|162.64 ± 8.43| 168.04 ± 13.31 | 176.18 ± 13.19 | 172.83 ± 6.30 |
> > > | discovery| **9.95 ± 1.01** |6.55 ± 1.60|5.23 ± 0.56| 6.63 ± 0.32| 5.48 ± 2.06| 7.23 ± 1.09 |
> > > | dispersion| **1.90 ± 0.28** |1.38 ± 0.37|0.95±0.09| 1.38 ± 0.13|1.48± 0.12| 1.31 ± 0.34 |
> > > | joint\_passage| **2.53 ± 0.17** | 2.25 ± 0.37| 1.61 ± 0.22|2.00±0.53 |2.46±0.02| 2.20 ± 0.46 |
> > > | joint\_passage\_size| **5.11 ± 0.01** |4.90±0.10|4.88±0.10|5.02±0.03| 4.77 ± 0.10 | 4.77 ± 0.16 |
> > > | sampling| **334.74 ± 4.05**|315.34±13.61|291.96±23.05|281.45±25.89|311.32±7.54 | 317.39 ± 10.06 |
> > > | wheel| **-21.52 ± 0.71** |-24.48 ± 0.33|-25.95±1.59|-25.12±0.93|-22.54±0.00| -24.50 ± 1.25 |
> > >
> > > Our extended results on more heterogeneous VMAS scenarios show that **partial parameter-sharing methods can indeed be advantageous in certain settings**, while **HPS maintains strong and stable performance across a wider range of environments**. This indicates that our experimental setup is sufficiently diverse to reflect these differences.
> > >
> > > These findings provide a more complete picture: instead of favoring a specific design, the benchmarks capture how methods behave under different conflict structures, while **HPS offers a robust alternative that adapts well without requiring explicit partial-sharing mechanisms**.

---

### Official Review · Reviewer_un9m · 2026-03-10

**Soundness:** 3
**Presentation:** 3
**Significance:** 2
**Originality:** 2
**Overall Recommendation:** 4
**Confidence:** 4

**Summary:**

This paper investigates the lack of behavioral diversity in Multi-Agent Reinforcement Learning (MARL) resulting from parameter sharing. The authors propose Geometric Gradient Decomposition Analysis, decomposing shared parameter gradients into orthogonal radial (magnitude) and tangential (direction) components. This analysis reveals an insight: while agents maintain high consistency in directional updates, they suffer from severe conflicts in magnitude updates. Consequently, the authors introduce Hyperspherical Parameter Sharing (HPS), featuring two core modules: (1) a Riemannian Backbone that constrains shared weights to a unit hypersphere to isolate directional learning and eliminate magnitude interference via geometric gradient filtering; and (2) an Agent-Specific Magnitude Generator that employs a FiLM modulation mechanism to provide individual scaling factors, restoring heterogeneous response intensities. Evaluated on SMAC, SMACv2, VMAS, and Predator-Prey, HPS consistently outperforms state-of-the-art methods across most scenarios.

**Compliance With Llm Reviewing Policy:**

Affirmed.

**Key Questions For Authors:**

1. The hyperspherical projection in HPS is formally very similar to Weight Normalization (Salimans & Kingma, 2016). As you both normalize the weights and then introduce a learnable scale parameter. What do you consider to be the fundamental difference between HPS and Weight Normalization? Have you conducted direct comparative experiments (e.g., replacing your approach with standard Weight Normalization + an agent-specific learnable scalar)?

2. The geometric gradient decomposition analysis (Figure 1) is conducted on fully connected layers. Do the distributions of magnitude and directional conflicts in the GRU layers exhibit the same pattern? If magnitude conflicts also dominate in the GRU layers, would the absence of Riemannian constraints on them become a performance bottleneck?

3. The auto-masking behavior (where the magnitude coefficients of dead agents approach zero in Figure 7) is an interesting emergent property. Is this behavior stable and reliable? Can it be consistently observed across all scenarios and seeds? If the magnitudes of dead agents do not drop to zero in certain cases, would this introduce noise and degrade performance?

4. Can your method be extended to policy gradient frameworks (such as MAPPO)?

**Limitations:**

The author should include more details on the comparison with weight normalization method.

**Strengths And Weaknesses:**

Strength:
1. Decomposing gradient conflicts into directional and magnitude dimensions provides a novel and intuitively sound perspective. The empirical analysis (Figure 1) demonstrates greate performance across multiple scenarios, providing a solid motivation for the proposed method.

2. The hyperspherical projection naturally eliminates magnitude gradients (Eq. 9, 12), achieving a provably structural decoupling of direction and magnitude. This approach is mathematically more rigorous than prior methods (e.g., the scalar conflict metric in GradPS). Furthermore, the geometric isolation theorem (Section 4.3) is concise and compelling.

3. The evaluation spans discrete control (SMAC/SMACv2), continuous control (VMAS), and classical coordination games (Predator-Prey), utilizing three different mixing networks (VDN/QMIX/QPLEX). The ablation studies and interpretability analyses (Figures 7 and 12) are sufficiently comprehensive.

Weakness:
1. Even the authors claimed that HPS is a parameter efficient method comparing to other branching-based method. HPS still requires 254% more parameters than FuPS in small scenarios (5m_vs_6m). This trade-off must be discussed much more transparently in the main text.

2. The conclusion admits performance drops in environments requiring strong sharing. The authors should investigate adaptive directional sharing mechanisms or provide experiments isolating the failure cases.

3. In the paper, the authors only applied magnitude modulation at the GRU output rather than applying Riemannian constraints within the GRU itself.

4. The core HPS projection seems to be a conditional form of Weight Normalization (Salimans & Kingma, 2016).

5. All experiments rely on value decomposition frameworks (VDN/QMIX/QPLEX). Validating HPS on Policy Gradient architectures (e.g., MAPPO) is critical to prove the method's claimed generality.

---

> ### Author Rebuttal · Authors · 2026-03-31
>
> We sincerely thank for the detailed review and the recognition of our novel gradient decomposition perspective, mathematical rigor of the geometric isolation theorem, and comprehensive evaluation.
>
> ## W1: HPS requires 254% parameters vs FuPS in small scenarios
> We will clarify this explicitly in the main text. The intended point is that HPS offers a middle ground: more expressive than fully shared baselines, while still scaling more favorably than branching-based approaches as the number of agents grows.
>
> ## W2: Performance drops in environments requiring strong sharing
>
> Although HPS effectively reduces the dominant **scale conflict** and improves **early-stage optimization**, in late-stage training it yields a much higher **Dormant Neuron ratio** than fully shared QMIX (e.g., **13.67% vs. 4.05%** on **zerg_5_vs_5**, **19.77% vs. 8.74%** on **zerg_5_vs_6**, $\tau=0.3$), which likely makes representation learning more fragile. In contrast, full sharing seems to provide stronger **implicit regularization** in such regimes.
>
> ## W3: Only applied modulation at GRU output, not within GRU
>
> See Q2 below.
>
> ## W4: Core HPS projection seems to be conditional Weight Normalization
>
> See Q1 below.
>
> ## W5: No MAPPO validation
>
> See Q4 below.
>
> ## Q1: Fundamental difference between HPS and WN
>
> HPS is not a straightforward conditional WN design. Unlike the standard MARL practice of feeding **obs+ID** into the shared backbone, HPS uses **observations only** in the backbone and reserves agent identity for the **Scale Generator**, since injecting ID into the backbone weakens the intended direction-scale decoupling. HPS also uses **residual scaling** $1+\gamma$ with zero initialization of the final Scale Generator layer, rather than direct FiLM-style gating \(\gamma\).
>
> Following the suggestion, we added a direct **WN + Agent Scale** baseline, where the HPS backbone is replaced by **standard Weight Normalization** and its output features are modulated by an **agent-specific learnable scalar**. It still remains below HPS, indicating that the gain comes from the proposed **direction-scale decoupling** and its more refined **input-routing/scaling design**, rather than a simple WN + scaling combination.
>
> | Scenario | Method | Result |
> |-|---|---|
> |5m_vs_6m|HPS|**72.66±9.22**|
> ||WN+Agent Scale|68.36±8.34|
> |MMM2|HPS|**85.35±3.87**|
> ||WN+Agent Scale|49.02±20.90|
>
> ## Q2: Do GRU layers exhibit the same gradient conflict pattern?
>
> Yes. We extended the analysis to the **GRU layers** (hh/ih × new/reset/update) on **10m_vs_11m, 5m_vs_6m, and MMM2**, and found the same qualitative pattern as in Figure 1: in **all tested GRU components**, **parallel/scale-related conflict is larger than perpendicular/direction-related conflict**. This shows that the scale-dominant conflict structure is not limited to fully connected layers.
>
> We apply modulation only at the **GRU output** as a **minimal, controlled intervention**. Imposing Riemannian constraints inside GRU would alter recurrent transitions and gate dynamics, introducing additional optimization/stability issues beyond the current scope. We therefore leave GRU-internal geometric constraints to future work.
>
> | Scenario|scale| direction|
> |----------------------|----------|---------|
> |hh_new/10m_vs_11m|45.98±2.89|8.63±4.76|
> |hh_new/5m_vs_6m|58.75±3.82|7.11±7.97|
> |hh_new/MMM2|45.27±2.12|9.69±3.76|
> |hh_reset/10m_vs_11m|46.91±2.81|17.03±5.67|
> |hh_reset/5m_vs_6m|62.89±3.67|17.58±9.63|
> |hh_reset/MMM2|53.55±2.71|16.76±3.29|
> |hh_update/10m_vs_11m|64.49±3.18|24.77±7.99|
> |hh_update/5m_vs_6m|67.27±4.34|22.73±8.01|
> |hh_update/MMM2|70.39±5.72|36.68±5.05|
> |ih_new/10m_vs_11m|53.09±5.20|13.98±1.44|
> |ih_new/5m_vs_6m|46.09±6.91|5.16±6.06|
> |ih_new/MMM2|50.08±2.59|10.82±3.26|
> |ih_reset/10m_vs_11m|56.72±2.90|18.52±2.47|
> |ih_reset/5m_vs_6m|64.22±6.36|20.47±9.24|
> |ih_reset/MMM2|60.94±3.33|18.24±4.09|
> |ih_update/10m_vs_11m|64.10±4.49|32.66±4.62|
> |ih_update/5m_vs_6m|68.91±7.08|25.39±7.68|
> |ih_update/MMM2|64.49±1.97|28.36±7.47|
>
> ## Q3: Stability of auto-masking behavior
>
> We verified the auto-masking behavior across scenarios and seeds(https://anonymous.4open.science/r/data-3F89/selected_dims_all_scenarios.png). The effect is stable: in most tested settings, dead agents consistently receive much smaller magnitude than alive agents. Even when the dead-agent scale does not fully vanish, it remains clearly lower than that of active agents, which still serves an isolation-like role and suppresses its influence rather than introducing obvious noise.
>
> ## Q4: Extension to MAPPO
>
> We added experiments on **MAPPO**. The results show that HPS remains effective beyond value decomposition: on **5m_vs_6m**, performance improves from **85.50 ± 1.62** to **88.84 ± 0.21**; on **3s5z_vs_3s6z**, from **82.52 ± 39.48** to **87.05 ± 29.01**.
>
> |Scenario|Method|Result|
> |----------------|-----|----------|
> | 5m_vs_6m| MAPPO|85.50±1.62|
> ||HPS|88.84±0.21|
> |MMM2|MAPPO |91.45±7.49|
> ||HPS|91.37±1.67|
> |3s5z_vs_3s6z|MAPPO|82.52±39.48|
> ||HPS|87.05±29.01|

---

> > ### Author Rebuttal · Reviewer_un9m · 2026-04-01
> >
> > Thank the authors for their detailed response. I have a few other follow-up questions I would like to add.
> > 1. How did the authors jump to the conclusion that 'in late-stage training it yields a much higher Dormant Neuron ratio than fully shared QMIX'. I believe a heatmap on the neural network would help to clarify.
> > 2. Could you elaborate more on the implementation on the GRU. I understand the first 5000 words might not be able to fully addressed the implementation. It would be helpful to address the implementation of the HPS on GRU in details.
> > 3. The improvement on MAPPO seems not quite good as the simple baselines. Is it because the MAPPO's implementation already has the HPS idea inherently? Since it has different actor and shared value function to handle the different actor's gradient.

---

> > > ### Author Response · Authors · 2026-04-02
> > >
> > > ## Q1: heatmap on the neural network
> > > We thank the reviewer for this helpful suggestion. To better support this point, we analyzed the dormant-neuron behavior at four checkpoints. For each checkpoint, we ran **10 episodes** on **zerg_5_vs_6** and measured the dormant-neuron ratio.
> > >
> > > Following the reviewer’s suggestion, we further visualize the **fc1 activations** of each agent as **heatmaps** at different training stages(https://anonymous.4open.science/r/data-3F89/10gen_zerg_dormant.png). The heatmaps show the same qualitative pattern: from both the per-agent and global perspectives, the proportion of inactive neurons increases in late training under HPS, and is more pronounced than in QMIX. We will add these quantitative statistics and heatmaps to make this conclusion better supported and more transparent. The results show a clear overall trend: the dormant-neuron ratio first decreases in early training, but then increases again in the later stage, which suggests a loss of representation capacity over time. Compared with fully shared **QMIX**, HPS exhibits a consistently higher dormant-neuron ratio, both **per agent** and **overall**, at late-stage checkpoints. This is why we hypothesize that dormant-neuron growth is a plausible contributor to the later-stage performance drop of HPS in these strong-sharing scenarios.
> > >
> > > ## Q2:the implementation on the GRU
> > > We thank the reviewer for this question. In the current paper, HPS is applied to the **GRU output** as a **minimal and controlled intervention**: the recurrent dynamics themselves are kept unchanged, and the hidden feature produced by the GRU is modulated by the agent-specific scale predicted by the Scale Generator. Concretely, the GRU computes the shared directional feature as usual, and HPS applies agent-conditioned residual scaling to this output feature, rather than modifying the recurrent gates or transitions themselves.
> > >
> > > We agree that more implementation detail on GRU is helpful, and we will clarify this in the revision. We also explored a more aggressive **GRU-internal direction-scale decoupling** variant. Specifically, we rewrote the GRU implementation by replacing all recurrent parameters (**weight_ih** and **weight_hh**) with the same hyperspherical projection used in the Riemannian Backbone, and fed the scale parameters produced by the Scale Generator into the GRU forward computation so that the scale modulation directly participated in the recurrent update.
> > >
> > > Empirically(https://anonymous.4open.science/r/data-3F89/hps_vs_gru.png), this GRU-internal variant improved performance noticeably in the early stage of training, but showed poor late-stage convergence and exhibits relatively large variance. We believe this may stem from multiple factors: (1) a single Scale Generator must simultaneously learn several different scale parameters for different recurrent matrices/gates, which makes optimization harder; and (2) imposing such decoupling inside the GRU may interfere with the recurrent transition dynamics and gate behavior, making memory evolution less stable. For this reason, we use output-side modulation in the current paper as the most stable and interpretable implementation of the core idea, while leaving GRU-internal geometric decoupling as an important future direction.
> > >
> > >
> > > | Scenario | Method | Win Rate|
> > > |---|---|---:|
> > > | 5m_vs_6m | HPS | 72.66% ± 9.22% |
> > > | 5m_vs_6m | HPS-GRU | 62.50% ± 14.95% |
> > > | MMM2 | HPS | 85.35% ± 3.87% |
> > > | MMM2 | HPS-GRU | 80.37% ± 18.57% |
> > >
> > > ## Q3: the performance of HPS in MAPPO
> > > We thank the reviewer for this insightful question. We would like to clarify that standard MAPPO still uses **full parameter sharing in the actor**: all agents share a single actor network, typically distinguished only by agent identity input. Therefore, the shared actor can still suffer from inter-agent gradient interference, and MAPPO does not inherently realize the HPS idea.
> > >
> > > The smaller improvement of HPS on MAPPO, compared with QMIX, is likely due to several architectural/optimization mechanisms already present in MAPPO. In particular, MAPPO commonly uses **LayerNorm**, **PPO clipping**, and **advantage normalization**, all of which can partially reduce gradient-scale disparity and stabilize shared optimization. As a result, the remaining headroom for explicitly resolving scale conflict may be smaller than in value-based methods such as QMIX, where raw TD-error gradients directly update shared parameters.
> > >
> > > That said, HPS still brings consistent gains on MAPPO because its benefit is not simply generic normalization. Instead, HPS provides **agent-specific dynamic scale modulation** and explicitly decouples shared directional learning from agent-specific scale variation, which cannot be fully achieved by normalization, clipping, or advantage normalization alone.

---

### Official Review · Reviewer_Dd7Z · 2026-03-11

**Soundness:** 3
**Presentation:** 3
**Significance:** 3
**Originality:** 2
**Overall Recommendation:** 4
**Confidence:** 4

**Summary:**

This paper studies gradient conflicts in parameter sharing for multi-agent reinforcement learning (MARL). The authors argue that existing approaches measure conflicts using scalar similarity metrics, which mix two different components of parameter updates: direction and scale. To analyze this issue, they introduce a geometric gradient decomposition that separates gradients into radial (scale) and tangential (direction) components. Based on the observation that agents tend to agree on direction but disagree on scale, the paper proposes Hyperspherical Parameter Sharing (HPS), which constrains shared backbone weights to a unit hypersphere and introduces an agent-specific scale generator. Experiments on several MARL benchmarks show improvements over existing parameter sharing baselines.

**Compliance With Llm Reviewing Policy:**

Affirmed.

**Key Questions For Authors:**

1. The analysis in Section 3 is very interesting and provides an insightful observation about gradient behavior in MARL. The results suggest that agents tend to agree on directional updates while disagreeing on scale updates. However, it would be helpful to better understand how general this phenomenon is across tasks. In SMAC, for example, there are many scenarios with different levels of heterogeneity and coordination difficulty. Did the authors observe consistent scale–direction patterns across different scenarios, or were there cases where the trend did not hold? Additionally, were there scenarios where the gap between scale conflicts and direction conflicts became particularly large or small? Further discussion on whether this behavior varies depending on the scenario characteristics would help clarify the generality of this finding.

2. How does HPS compare to simpler normalization or modulation baselines? It would be helpful to include comparisons with simpler baselines such as weight normalization or hyperspherical parameterization without the scale generator, or standard FiLM-style modulation applied to a shared backbone. This would help clarify which component of the proposed architecture contributes most to the performance gains.

3. What is the computational overhead of HPS compared to standard parameter sharing?
Although the paper discusses parameter counts, it would be helpful to report training time, memory overhead, or wall-clock training cost relative to baselines, especially for larger scenarios such as SMACv2.

**Limitations:**

Yes. The authors adequately discussed the limitations and potential negative societal impact.

**Strengths And Weaknesses:**

**Strenghts**

1. Interesting perspective on gradient conflicts

The paper introduces a geometric perspective on gradient conflicts in MARL parameter sharing by decomposing gradients into radial and tangential components. This provides an interpretable way to analyze optimization dynamics and offers a potentially useful diagnostic tool for understanding interference between agents. The empirical observation that directional gradients tend to be aligned while scale components often conflict is an interesting insight that may inspire future work.

2. Simple and intuitive architecture

The proposed HPS framework is conceptually simple and easy to implement. By constraining the backbone weights to the unit hypersphere and introducing an agent-specific scale generator, the architecture separates representation learning (direction) from response magnitude (scale). This design avoids more complex approaches such as network branching or role decomposition and maintains a shared backbone structure.

3. Empirical evaluation on several MARL benchmarks

The method is evaluated on multiple environments (SMAC, SMACv2, VMAS, Predator-Prey) and generally shows improved performance compared to several parameter sharing baselines.

**Weaknesses**

1. Limited novelty relative to existing normalization and modulation approaches

While the geometric interpretation is interesting, the proposed architecture resembles a combination of existing techniques such as weight normalization and feature-wise modulation (e.g., FiLM-style scaling). Constraining weights to lie on a hypersphere and learning multiplicative scaling factors is not fundamentally new in neural network design. The novelty therefore mainly lies in applying these ideas to MARL parameter sharing rather than introducing a fundamentally new learning principle.

2. Missing or weak baselines for isolating the source of improvements

The experiments do not include several baselines that would help isolate the contribution of the proposed components. For example, it would be helpful to compare against:

- simple weight normalization or hyperspherical constraints without scale generators
- FiLM-style modulation applied to standard parameter sharing networks
- alternative gradient conflict mitigation methods

Without these comparisons, it is difficult to determine whether the improvements come from the geometric decomposition insight or from more standard normalization/modulation effects.

3. Experimental design lacks consistency across sections

The experimental design appears somewhat fragmented across sections, which makes it difficult to interpret the results in a systematic way. For example, Section 5.3 evaluates the method only on VMAS environments, whereas Section 5.4 (ablation study) uses a different set of tasks: 5m vs 6m, MMM2 (SMACv1), and VMAS-sampling.
These choices appear somewhat ad hoc and make it unclear whether the conclusions generalize consistently across benchmarks. A more uniform experimental protocol (e.g., running ablations on the same benchmark suite used in the main experiments) would make the empirical evaluation more convincing.

4. Possible missing results in Figure 1

In Figure 1, some of the curves corresponding to certain thresholds appear to be missing. For example, in the 5m vs 6m plot, results for thresholds such as τ = 0.40 and τ = 0.50 do not seem to be shown, while other thresholds are plotted. It is unclear whether these curves were omitted intentionally or due to an error. Clarifying this would improve the reliability and clarity of the analysis presented in Section 3.

---

> ### Author Rebuttal · Authors · 2026-03-31
>
> We sincerely thank for the thoughtful review and the recognition of our geometric perspective on gradient conflicts, simple and intuitive architecture, and broad empirical evaluation.
>
> ## W1: Novelty
>
> We agree that HPS is related in form to Weight Normalization and FiLM-style scaling, and we do not claim these primitives are new in isolation. Our contribution is a **MARL-specific diagnosis and mechanism**: we show that parameter sharing is mainly hindered by **scale conflict**, and derive a structured decoupling between shared directional learning and agent-conditioned scale adjustment.
>
> HPS is not a direct WN+FiLM combination. It differs from **standard MARL architectures** in two key ways. **First, unlike the common practice of feeding concatenated obs+ID into the shared backbone**, HPS feeds the backbone with **observations only** and reserves agent ID exclusively for the **Scale Generator**; injecting ID into the backbone weakens the intended direction-scale decoupling and can even hurt performance. **Second, unlike standard FiLM-style modulation**, HPS uses **residual scaling** $1+\gamma$ with zero initialization of the final scale layer, so modulation starts from identity and gradually learns agent-specific scale corrections. These are structural differences tailored to clean direction-scale decoupling in MARL.
>
> ## W2: Simple baselines
>
> Following the recommendation, we added **WN-QMIX**, **FiLM-QMIX**, and **PCGrad-QMIX**. **WN-QMIX** is insufficient and can even underperform QMIX, while **FiLM-QMIX** captures part of the benefit but remains far below HPS on the harder heterogeneous **MMM2** task (**53.03 vs. 85.35**). **PCGrad-QMIX** also helps, yet remains consistently below HPS. Overall, these results suggest that HPS’s gain is not explained by normalization, generic modulation, or generic gradient surgery alone, but by the proposed **direction-scale decoupling**.
>
> |Scenario|Method|Result|
> |--------------|------|------|
> | 5m_vs_6m|HPS|72.66±9.22|
> ||QMIX|50.88±5.83|
> ||WN-QMIX|53.32±12.11|
> ||FILM-QMIX|65.78±15.59|
> ||PCGrad-QMIX|63.96±9.83|
> |MMM2|HPS|85.35±3.87|
> ||QMIX|47.27±25.28|
> ||WN-QMIX|30.08±16.94|
> ||FILM-QMIX|53.03±18.54|
> ||PCGrad-QMIX|64.75±23.97|
>
>
> ## W3: Experimental consistency
>
> We now expand the ablation to better align with the main benchmarks, and the new results support the same qualitative conclusions across the aligned task set.
>
> | Scenario|Algorithm|Raw|HPS|
> |------------------|---|---------|--------|
> | 5m_vs_6m|VDN|0.57±0.07|0.79±0.04|
> ||QMIX|0.51±0.06|0.73±0.09|
> ||QPLEX|0.68±0.07|0.75±0.04|
> | MMM2|VDN|0.02±0.01|0.09± 0.10|
> ||QMIX|0.47±0.25|0.85±0.04|
> ||QPLEX|0.50±0.25|0.85±0.04|
>
> ## W4: Possible missing results in Figure 1
>
> As noted in Sec. 3.2, some curves overlap because the number of agent pairs is very limited in small scenarios. For example, in **5m_vs_6m** there are only 5 agents, i.e., only $\binom{5}{2}=10$ pairs, so the conflict density can take only a small number of discrete values. As a result, thresholds $\tau \in \{0.35,\ldots,0.55\}$ cannot all produce visually distinct curves. We will improve the figure with clearer line styles, zoomed insets, and a numerical table for verification.
>
> ## Q1: Generality of scale–direction patterns across SMAC scenarios
>
> We extended the Sec. 3 analysis to additional SMAC scenarios. Across **all tested scenarios, both VDN/QMIX, and all thresholds**, **scale-related conflict is consistently larger than direction-related conflict**, with **no reversal observed**. The **gap is scenario-dependent**: it is larger on **5m_vs_6m, 8m_vs_9m, and MMM2**, and smaller on harder coordination-heavy scenarios such as **3s5z_vs_3s6z**, where both conflicts increase. On simpler maps such as **3s_vs_5z**, both conflicts are smaller overall, though the same qualitative trend still holds.
>
> |Scenario|Algorithm|Threshold|Scale|direction|
> |-----------------|----|---|---------|---------|
> |8m_vs_9m|QMIX|.35|53.09±0.74|33.67±0.78|
> |||.45|32.23±0.51|16.29±0.12|
> |||.50|32.23±0.51|9.18±0.35|
> | 10m_vs_11m|QMIX|.35|64.34±1.91|40.66±6.84|
> |||.45 |47.62±0.12|20.08±3.91|
> |||.50|29.10±0.12|11.52±0.90|
> | 3s_vs_5z|QMIX|.35|23.36±4.45|8.44± 1.48|
> |||.45|23.36±4.45|8.44±1.48|
> |||.50|23.36±4.45|8.44±1.48|
> | 3s5z_vs_3s6z|QMIX|.35|59.88±0.51|43.75±2.27|
> |||.45|36.76±0.20|23.63±0.90|
> |||.50|36.76±0.20|14.96±0.12|
>
>
> ## Q2: Comparison with simpler baselines
> see W2 above.
>
> ## Q3: Computational overhead of HPS
> We also provide measurements on an Intel i7-13700KF, NVIDIA RTX 3090, running each algorithm for **6.1M steps**:
>
> |Scenario|Method|GPU Memory(MB)|Training Time|
> |---------|---|------|--------|
> |protoss5X6|FuPS|689|3h 32min|
> ||FuPS+ID|747|3h 33min|
> ||**HPS**|**1119**|**3h52min**|
> | protoss10X11|FuPS|1038|5 09min|
> ||FuPS+ID|1186|5h 13min|
> ||**HPS**|**1574**|**5h47min**|
>
> HPS adds only **~20 min** overhead over FuPS on the larger scenario, and scales well with more agents.

---

> > ### Author Rebuttal · Reviewer_Dd7Z · 2026-04-03
> >
> > Thank you for the detailed and thoughtful rebuttal. I appreciate the additional experiments and clarifications, particularly the inclusion of stronger baselines (e.g., WN, FiLM, and PCGrad) and the extended analysis across more scenarios. These additions address several of my original concerns and strengthen the empirical evaluation of the paper.
> >
> > However, I still have some reservations regarding the overall novelty. While the authors clarify several architectural distinctions, these differences seem primarily at the implementation level, and it is not yet fully clear whether the proposed design constitutes a fundamentally new MARL-specific approach, rather than a careful combination of existing normalization and modulation techniques.
> >
> > Overall, I find the paper to be technically sound and the empirical results to be promising, but with some remaining limitations that affect its broader impact. I will therefore maintain my score.

---

### Decision · Program_Chairs · 2026-04-30

**Decision:**

Accept (regular)

**Comment:**

Based on the observation that agents tend to agree on direction but disagree on scale, this work proposes Hyperspherical Parameter Sharing (HPS), which constrains shared backbone weights to a unit hypersphere and introduces an agent-specific scale generator.

The strengths of this work are listed as follows:

1. The work offers a clear and useful perspective on gradient conflicts under parameter sharing by separating direction and scale.
2. The Geometric Gradient Decomposition Analysis part is quite good and easy to understand.
3. The experiments cover multiple benchmarks and compare against several representative baselines.

The major concerns of this work are:
1. HPS introduces additional parameters, and its advantages may be weaker in settings that require stronger sharing.
2. Some experiments do not fully reveal the bottlenecks of parameter sharing.
3. One reviewer still concerns with the experimental results, please describe in details regarding how other algorithms are evaluated and describe the difference among the setting among their original paper and this paper.

Overall, this work turns an empirical observation about the structure of shared gradients into a concrete method for MARL. Although some limitations remain, the work makes a valuable contribution to the MARL community and offers a meaningful perspective on the design of parameter-sharing methods. Therefore, I recommend acceptance.